# Impact of Holuhraun volcano aerosols on clouds in cloud-system resolving simulations

Mahnoosh Haghighatnasab [1], Jan Kretzschmar [1], Karoline Block [1], and Johannes Quaas [1]

[1]Institute for Meteorology, Universität Leipzig, Leipzig, Germany

**Correspondence:** Mahnoosh Haghighatnasab (Mahnoosh.Haghighatnasab@uni-leipzig.de)

**Abstract.** Increased anthropogenic aerosols result in an enhancement in cloud droplet number concentration ($N_d$), which consequently modifies the cloud and precipitation process. It is unclear how exactly cloud liquid water path (LWP) and cloud fraction respond to aerosol perturbations. A volcanic eruption may help to better understand and quantify the cloud response to external perturbations, with a focus on the short-term cloud adjustments. The goal of the present study is to understand and quantify the response of clouds to a selected volcanic eruption and to thereby advance the fundamental understanding of the cloud response to external forcing. In this study we used the ICON (ICOsahedral Non-hydrostatic) model in its numerical weather prediction setup at a cloud-system-resolving resolution of 2.5 km horizontally, to simulate the region around the Holuhraun volcano for one week (1 – 7 September 2014). A pair of simulations, with and without the volcanic aerosol plume, allowed us to assess the simulated effective radiative forcing and its mechanisms, as well as its impact on adjustments of LWP and cloud fraction to the perturbations of $N_d$. In comparison to MODIS (Moderate Resolution Imaging Spectroradiometer) satellite retrievals, a clear enhancement of $N_d$ due to the volcanic aerosol is detected and attributed. In contrast, no changes in either LWP or cloud fraction could be attributed. The on average almost unchanged LWP is a result of some LWP enhancement for thick, and a decrease for thin clouds.

## 1 Introduction

Volcanic eruptions influence the climate by emitting large quantities of solid particles (ash) and gaseous compounds into the atmosphere (Cole-Dai, 2010). Ash particles block sunlight and, therefore, decrease solar radiation reaching the surface. This leads to a cooling, even if the ash settles down due to gravity relatively fast (Robock, 1981).

The gas emissions mainly include water vapor, carbon dioxide, sulfur components (mainly sulfur dioxide ($SO_2$)), and nitrogen (Mather et al., 2004). Chemical processes convert $SO_2$ to sulfuric acid ($H_2SO_4$; sulfate aerosol) in the troposphere at relatively short time spans of few days, while in the stratosphere, the conversion can take weeks up to months (Rose et al., 2001).

Sulfate aerosols, injected from a large volcanic eruption, modify the Earth's radiative budget directly by scattering sunlight and indirectly via interaction with clouds (Sahyoun et al., 2019). The latter is the focus of the present manuscript. A large volcanic eruption as a natural laboratory may help to better understand and quantify how cloud properties are modified in response to anthropogenic aerosols emissions (Inguaggiato et al., 2018; Christensen et al., 2021).

Imposed effective radiative forcing by aerosol-cloud interactions in warm clouds can be separated into the Twomey effect (Twomey, 1974) and cloud adjustments to radiative forcing (Bellouin et al., 2020). An enhancement in cloud condensation nuclei (CCN) concentrations lead to an increase in cloud droplet number concentration ($N_d$), resulting in a smaller effective radius ($r_e$) if cloud liquid water path (LWP) is constant. Consequently, scattering cross section and the cloud albedo are enhanced, causing clouds to reflect more sunlight back to space, which is known as Twomey effect (Twomey, 1974). Anthropogenic aerosols modify cloud particle size distributions, which reduces the efficiency of collision-coalescence processes, leading to delay in precipitation onset consequently enhancing cloud lifetime (Albrecht, 1989). This infers on average an enhancement in cloud fraction and LWP (Pincus and Baker, 1994; Gryspeerdt et al., 2019). These longer lived clouds reflect more sunlight back to space and cool the atmosphere and surface even more, which is known as lifetime effect (Xue et al., 2008).

Along with the aforementioned effects, there is a large variety of processes that partially offset these effects on clouds, such as a reduced maximum supersaturation if more droplets compete for the available water vapor (Twomey, 1959), a larger evaporation rate of smaller droplets (Small et al., 2009), increased droplet spectrum dispersion (Brenguier et al., 2011; Liu and Daum, 2002), or enhanced evaporation due to cloud-top mixing (Ackerman et al., 2004; Gryspeerdt et al., 2019). Because the different effects oppose each other, the overall changes in the effective radiative forcing could be minor on larger scales (Khain et al., 2008; Stevens and Feingold, 2009). In this study, the responses of clouds to aerosols emitted in the Holuhraun volcano eruption were examined. The Holuhraun eruption was the strongest in Europe since the 18$^{th}$ century and emitted substantial amounts of sulfate aerosol (Ilyinskaya et al., 2017). This natural phenomenon has triggered a large effort to investigate the impact of this large aerosol perturbation on cloud properties. (Malavelle et al., 2017).

Malavelle et al. found a significant reduction in $r_e$ in satellite data, but only insignificant alterations of LWP. They further concluded that several general circulation models overemphasize LWP increase in response to the additional aerosol. However, McCoy et al. (2018) did find an increase in LWP when carefully conditioning on moisture convergence. In addition, ambiguous results, with LWP responses of either sign, were obtained by Toll et al. (2017) when analyzing multiple volcanic eruptions. Several cloud-resolving modeling studies on the sensitivity of LWP to variation in cloud droplet number concentration have been conducted. Bretherton et al. (2007) examined the effect of entrainment and sedimentation rate on LWP in stratocumulus cloud regimes using large eddy simulation (LES). Their results explained the process details of the conclusions by Ackerman et al. (2004), namely that sedimentation leads to decrease of entrainment rates and increase of LWP. Seifert et al. (2015) conducted a set of LES simulations over fair weather cumulus cloud regimes over the subtropical ocean. They concluded that in this cloud regime, the response of LWP on enhancing $N_d$ was almost negligible in equilibrium, and slight reduction in cloud cover was obtained, leading to a negative cloud lifetime effect, compensating the positive radiative forcing of the Twomey effect. Lebo and Feingold (2014) performed LES simulations of two different cloud regimes of marine stratocumulus and trade wind cumulus clouds. They showed different relationships between relative LWP response to relative change of aerosol concentration $N_a$, a term the called $\lambda$, and the precipitation probability susceptibility ($S_{POP}$). For trade wind cumulus clouds regime, $\lambda$ decreases with enhancement of $S_{POP}$, because of entrainment and evaporation rate effect in cumulus clouds. In stratocumulus clouds, $\lambda$ and $S_{POP}$, in contrast, were positively related. In this case, aerosol-induced evaporation–entrainment and/or sedimentation–entrainment effects restricted further increase in LWP in their simulations. Sato et al. (2018) conducted one-year global

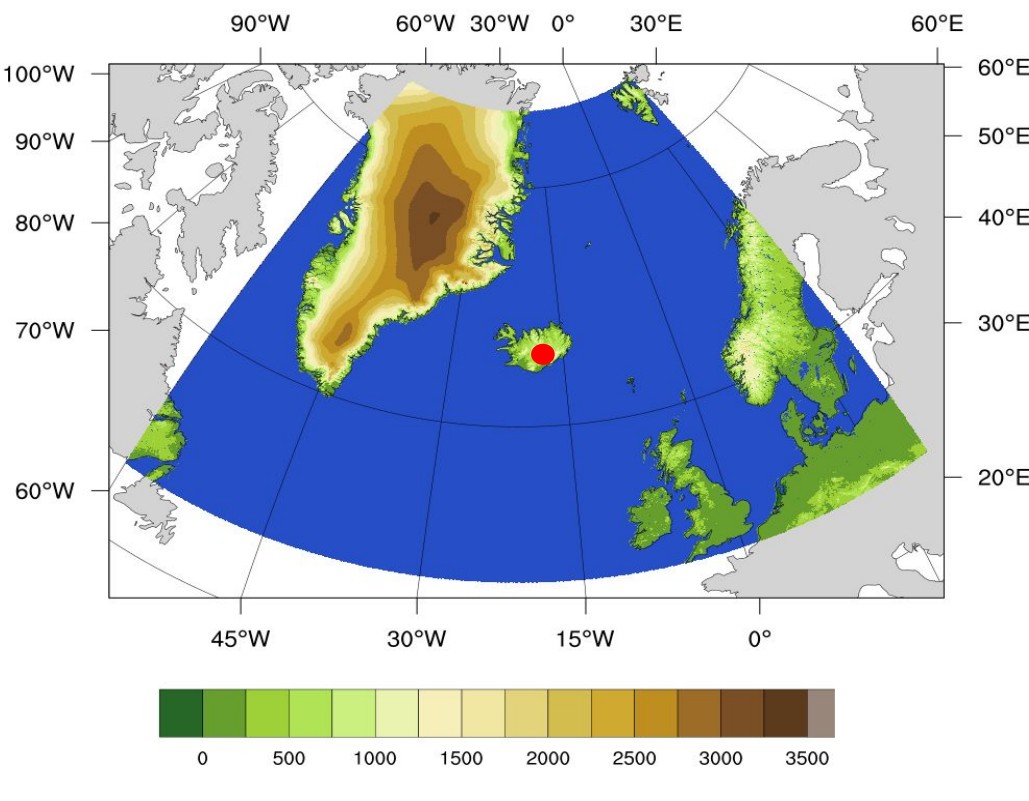

**Figure 1.** Domain of the ICON-NWP simulations over the North Atlantic ocean (60°W - 20°E, 50°N - 80°N) which included the Holuhraun volcano on Iceland that erupted in September 2014. The model resolution is approximately 2.5 km in the horizontal (R2B10 triangular grid). Red dot indicates location of Holuhraun volcano. Blue color indicates the ocean and color bar indicates the elevation of land above sea surface (in m) in ICON-NWP model.

cloud-resolving simulation to examine the sensitivity of liquid water content (LWC) to aerosol loading. They demonstrated that in their model, the condensation process in the lower part of clouds is associated with positive LWC response and evaporation process in upper part of clouds is responsible for negative response to additional aerosols loading. Following these previous studies, we chose the Holuhraun eruption to investigate the response of LWP, cloud fraction, and its corresponding radiative effect in response to additional CCN in the emission plume of the volcano, employing simulations in cloud resolving resolution and comparing them to satellite observations.

## 2 Model and data

The present study focuses on a detection and attribution approach, using cloud resolving simulations (kilometer-scale resolution, Stevens et al., 2020) in combination with the analysis of satellite data. A pair of simulations over the North Atlantic ocean around the Holuhraun volcano on Iceland was employed (Figure 1). The model used is the ICOsahedral Non-hydrostatic model (ICON, Zängl et al., 2015). The ICON model is developed by a collaboration between the German Meteorological Service and the Max Plank Institute for Meteorology (Klocke et al., 2017). It can be used from global simulation in the climate scale (Giorgetta et al., 2018) to high resolution large eddy simulations (Heinze et al., 2017). Here, the physics package of the numerical weather prediction (NWP) variant is used (ICON-NWP). The resolution corresponds to approximately 2.5 km in the horizontal (R2B10 triangular grid). Vertically, 75 layers with top height at 30 km were chosen. The vertical resolution increases towards model top with a model layer thickness of 20 m in the boundary layer and a maximum layer thickens of 400 m near model top.

The physics package of ICON-NWP includes a comprehensive double moment cloud liquid and ice microphysical scheme (Seifert and Beheng, 2006). Because of using a rather fine resolution, deep convection is considered to be resolved, whereas, for shallow convection, the parameterization scheme by Tiedtke (1989) with modifications by Bechtold et al. (2008) was used. A grid-scale cloud cover scheme was employed in simulations. In this scheme, if the sum of specific cloud water content and specific cloud ice content is larger than a certain threshold, cloud fraction is set to 1 and else set to 0, and the Tiedtke (1989) shallow convection scheme contributes to the computation of specific cloud water and ice content. To achieve a more realistic representation of the Twomey effect, we furthermore coupled the hydrometeor number concentrations from the double moment microphysical scheme to the radiation scheme as proposed in Kretzschmar et al. (2020).

Initial and boundary conditions were derived from the European Centre for Medium-Range Weather Forecast (ECMWF) Integrated Forecasting System (IFS) operational analysis. Variables such as temperature, wind, geopotential, humidity and hydrometeors were used in initial and boundary conditions. The 2014 Holuhraun eruption was a fissure eruption that started on 20 August 2014 and ended on 25 February 2015. By 7 September 2014, the lava field had extended more than 11 km to the north (Kolzenburg et al., 2017). We choose the period from 1 to 7 September 2014 for our analysis because the lava field had developed sufficiently in this period and substantial amounts of $SO_2$ had been emitted into the atmosphere, while, at the same time, a well-defined plume is observable. The first 9 hours of the simulations were excluded from analyses so that the spin-up effect is sufficiently small in our simulations. An additional feature in simulations that must be mentioned, is the implementation of a satellite simulator into the model. Satellites are essential tools to assess the character of clouds due to their global coverage and availability (Lai et al., 2019). Differences between model simulations and satellite retrievals stem in part from a different definition of the respective quantities that are compared. Therefore, one approach to reduce inconstancy between model simulations and satellite retrievals is to use satellite simulators in models to mimic the observational processes (Roh et al., 2020). The COSP satellite simulator (Bodas-Salcedo et al., 2011) is an open source work package developed by CFMIP (Cloud Feedback Model Intercomparison Project) to replicate active and passive satellite data using variables from the model as an input (Webb et al., 2017). In this study, just satellite simulator for MODIS (Moderate Resolution Imaging Spectroradiometer) observations (Pincus et al., 2012) was used. The COSP simulator uses several model variables as

input such as temperature, pressure, cloud fraction and cloud water content (Kretzschmar et al., 2019) to generate what the MODIS retrievals would capture given the simulated clouds fields (Saponaro et al., 2020). In addition of above mentioned variables, effective radius of liquid cloud droplets and ice crystals are considered as MODIS simulator's input. Effective radius of cloud droplets and ice crystals were calculated from parameters derived from size distribution function of hydrometeor in two-moment microphysic scheme. The satellite simulator has previously been implemented and used in ICON-NWP by Kretzschmar et al. (2020).

In the cloud-resolving simulation (each grid box is either fully cloudy or clear), the use of sub-grid variability, one of the features of COSP for application in general circulation models, was not necessary. In order to evaluate COSP related variables in our simulations, the collection 6.1 Level-2 MODIS-Aqua optical and physical cloud data product was used (Platnick et al., 2017); therefore, swaths with 1 km spatial resolution for $r_e$, cloud optical thickness ($\tau_c$), LWP along with cloud fraction with 5 km spatial resolution were used and remapped to the model resolution to have an accurate comparison. To remap the MODIS granule to a latitude / longitude grid, for each specific point the mean value of each variable in each swath is computed. It should be mentioned that even using a satellite simulator there is an uncertainty between definition of LWP in simulation and MODIS observations, because the bulk microphysic scheme has gap in size distribution between cloud droplets and rain that is not necessarily the same as in the visible/near-infrared retrievals by MODIS. To a lesser extent, this issue may also affect the computation of $N_d$. Furthermore, the planetary albedo at the top of the atmosphere is analyzed as retrieved by the Clouds and the Earth's Radiant Energy System (CERES) instrument on board the Aqua satellite (Su et al., 2015; Loeb et al., 2016).

## 2.1 CCN activation

The ICON-NWP version applied in this study does not contain an interactive aerosol model; therefore, in this section, we discuss how CCN are activated into clouds droplets in the default model setup and afterward we introduce a new method for CCN activation in microphysical scheme, which had specifically been developed for this study. In the default setup of ICON, CCN activation uses a parameterization that employs a functional dependency of grid scale vertical velocity and pressure to derive the number of newly activated CCN (Hande et al., 2016). Hande et al. (2016) performed model simulations that considered a multi-modal interactive aerosol scheme to provide information on the formation and transport of aerosols in Europe and, by using the parameterization of Abdul-Razzak and Ghan (2000, ARG), derived CCN number concentrations for different vertical velocities for a selected date (30 April 2013). This parameterization thus assumes a temporally and spatially constant profile of CCN which is representative for CCN background over Europe. For that reason, this parameterization alone can not provide information about CCN concentration within a plume of volcanic aerosol.

In order to more accurately represent the aerosol plume, we use look-up tables that contain the number of activated CCN as a function of pressure $p$ and vertical velocity $w$ as an input for the ICON simulation. The number of activated CCN is interpolated from these look-up tables considering the values of $p$ and $w$ in each grid-box within the cloud microphysical scheme. This method had been developed for the ICON model in its large-eddy setup (Costa-Surós et al., 2020) and has been implemented into ICON-NWP for our study. While dedicated interactive-aerosol simulations were performed to create the look-up tables in Costa-Surós et al. (2020), we use the Copernicus Atmospheric Monitoring Service (CAMS) reanalysis (Inness et al., 2019)

to obtain the information about the spatial-temporal distribution of the aerosol mass mixing ratio by aerosol species. The CAMS reanalysis provides aerosol mass mixing ratios at 60 hybrid sigma/ pressure levels up to 0.1 hPa, and covers the 2003 to 2020 period. Using the aerosol mass mixing ratio from the CAMS reanalysis, along with using the ARG parametrization, that calculates the number of activated aerosols employing the Köhler theory (Köhler, 1936), we created look-up tables of activated CCN for our simulation.

In our study, the ARG-parameterization is employed offline, by running a box model setup and using aerosol mass mixing ratio from the CAMS reanalyses as an input for various vertical velocities. The ARG-parameterization has been used in microphysical schemes in a wide range of resolutions before, ranging from global climate models to cloud resolving models (Ghan et al., 2011; West et al., 2014; Luo et al., 2008; Goto et al., 2020). The ARG parameterization is based on the competition between aerosol particles for available water vapor which depends on aerosol particle composition, size distribution

and most importantly the supersaturation forcing rate obtained by the updraft. We evaluate supersaturation $S_{0,i}$ at ten specific values of vertical velocity used in the look-up tables (see Costa-Surós et al., 2020). After calculating $S_{\mathrm{max}}$, the critical radius of activation for each aerosol mode is obtained in the box model. When the supersaturation for each aerosol mode is smaller than maximum supersaturation $S_{\mathrm{max}} \geq S_{0,i}$ , the environment has gained the needed supersaturation to activate the particles. Using this approach, an observations-tied spatially-temporally varying input number concentration of activated CCN for ten

prescribed vertical velocity classes was produced. In the CAMS reanalyses data, the aerosols emitted from the Holuhraun volcano are not represented: it is firstly not constrained by the data assimilation. CAMS assimilates MODIS aerosol optical depth (AOD) retrievals (Levy et al., 2013), but due to the presence of extensive clouds in the region of interest, MODIS was not able to capture sufficient information about AOD. Secondly, it is also not included in the model simulation, because in the emission source model of CAMS, no volcanic emissions are considered. Therefore, the CAMS data was used to obtain

background spatial and temporal aerosols concentration and in order to implement aerosol concentrations inside the plume, the sulfate aerosol concentration in CAMS was scaled based on the $SO_2$ emission monitored by Ozone Mapping and Profile Suite (OMPS) satellite retrievals which will be explained in more detail in the next session (Yang, 2017). The approach used in this study does not use an interactive aerosol physics which would simulate the evolution of aerosol field by transport and transformation. However, an important buffering mechanism, namely the consumption of CCN by activation, is considered in

this study (Costa-Surós et al., 2020). So the CCN are depleted when they are activated and thereafter are relaxed back to their initial profile. This is implemented by a simple prognostic equation for the CCN concentration that considers a sink for CCN at droplet activation and a source by relaxation to the prescribed CCN profile, advection is not computed. It should also be noted that there is an important advantage of our method compared to a fully interactive aerosol scheme, which is that the location of the plume is derived from observations and therefore is at the same region as in satellite retrievals. This allowed us to analyze

inside and outside of the plume in simulations and satellite products with confidence.

## 2.2    The volcanic-aerosol plume in the model simulations

Lava flows and emitted gases from volcanic eruptions are the most common features that remotely can be monitored globally and at different time scales. $SO_2$ is one of the most common gases emitted from volcanic eruptions and can be retrieved by

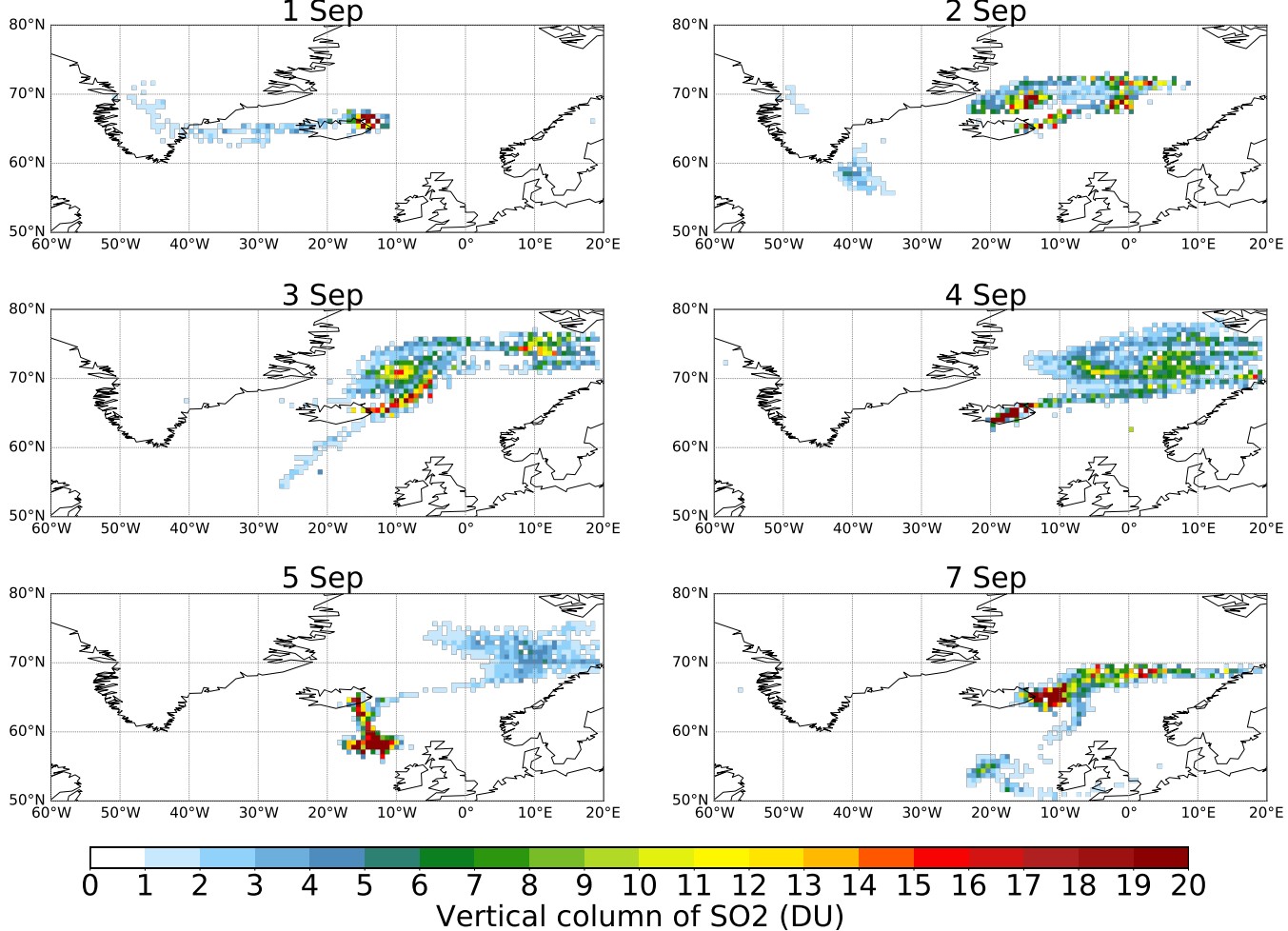

**Figure 2.** Total vertical column amount of SO₂ associated with the ground pixel retrieved using a prescribed SO₂ profile centered at 3 km (in Dobson units) from 1 to 7 September 2014 obtained from OMPS (Yang, 2017) satellite retrievals. No data are available for 6 September 2014.

spaced-based sensors (Fioletov et al., 2020). In this study, the OMPS data product (Level 2) for SO₂ was used. This data set provides information about vertically integrated SO₂ (in Dobson units, DU). The SO₂ retrievals for 1 to 7 September 2014 for the lower troposphere are shown in Figure 2.

The SO₂ plume was detected on 1 September shortly after the beginning of the eruption and evolved over time mostly east-wards, towards Scandinavia. Former studies compared OMPS satellite retrievals with surface observations for the Holuhraun eruption and showed that satellite retrievals are able to detect spatial and temporal evolution of the volcanic plume (Ialongo et al., 2015). In this study, we performed two simulations over the domain shown in Figure 1, one with background aerosol concentrations only, which is referred to as the no-volcano simulation, and one with scaling the sulfate concentrations in the

CAMS reanalysis data within the plume as defined by the OMPS satellite retrievals, referred to as the volcano simulation in this article. As shown in Figure 2, grid-points with $SO_2$ concentrations in the lower troposphere exceeding 1 DU are considered to constitute the plume. For these grid-points, a scale factor field was computed by dividing the $SO_2$ concentrations retrieved within the plume by the mean $SO_2$ concentration for the entire domain outside the plume region. In the next step, the sulfate aerosol mass mixing ratio from the CAMS reanalyses was scaled inside of plume by these scaling factors to derive a new CCN distribution that now considers the volcanic plume with the enhancement consistent with the OMPS satellite retrievals. $SO_2$ is considered a proxy of the loading of additional sulfate aerosols in a volcanic plume. The potentially activated CCN concentration was computed from vertically-resolved aerosol components (including sulfate) mass mixing ratio using a box model. The potentially activated CCN profile that is produced to be used as input in ICON-NWP is thus also resolved in vertical levels. In order to define the volcanic plume on the basis of the distribution of sulfate aerosol from the CAMS reanalysis, we scaled each vertical level of sulfate aerosol in CAMS based on the lower troposphere (up to 3 km) column amount of $SO_2$ in OMPS data. In consequence, the vertical distribution within the plume follows the one generated by the reanalysis without the plume, but the scaling makes use of the vertical information from the satellite retrievals in such that only the boundary-layer enhancement is used, i.e. the aerosol that is relevant for the formation of the liquid water clouds investigated in our study. It should be mentioned that in this study, the emission of water vapor from volcanic eruption hasn't been taken to account.

Figure 3 shows the geographical distribution of vertical-mean number of activated CCN for 2 September 2014 with a background sulfate aerosol concentration (a and c) and scaled sulfate concentration (b and d) for two different prescribed vertical velocities ($0.599\,\mathrm{m\,s^{-1}}$ and $4.64\,\mathrm{m\,s^{-1}}$). As is mentioned in section 2.1, the strength of the updraft corresponds to maximum supersaturation in the ARG-parameterization. Therefore, more CCN gets activated at higher vertical velocity. In Figure 3, the location of the plume can smoothly be identified. This information lead us to perform two simulations one with a background activated CCN concentration (left panels) referred as no-volcano simulation, and one with scaled activated CCN concentration (right panels) referred to as volcano simulation.

## 3  Results

The present study aims at a detection and attribution approach, assessing the differences in cloud properties within and outside the volcanic plume by comparing simulations with satellite observations. It has been shown that meteorological conditions and cloud regimes are important to determine the effect of additional aerosol loading on cloud microphysical properties. Figure 4 indicates visible image obtained from MODIS-AQUA satellite retrievals. A synoptic frontal system is located over the North Atlantic ocean and contains large-scale, mostly stratiform ice and liquid phase clouds. These conditions remain similar during the simulation period. In order to to select liquid phased clouds in the MODIS data, the Cloud Phase Optical Properties flag was used. For simulations, the COSP simulator produces the microphysical properties for the liquid and ice phase clouds separately and we used only the outputs dedicated to liquid phase clouds in our analyses. This study aims to evaluate how cloud microphysical properties ($N_\mathrm{d}$ and LWP) behave differently in and outside the volcano plume. To address this scientific question, grid cells that are located inside and outside of volcano plume are analyzed and compared to each other in volcano and

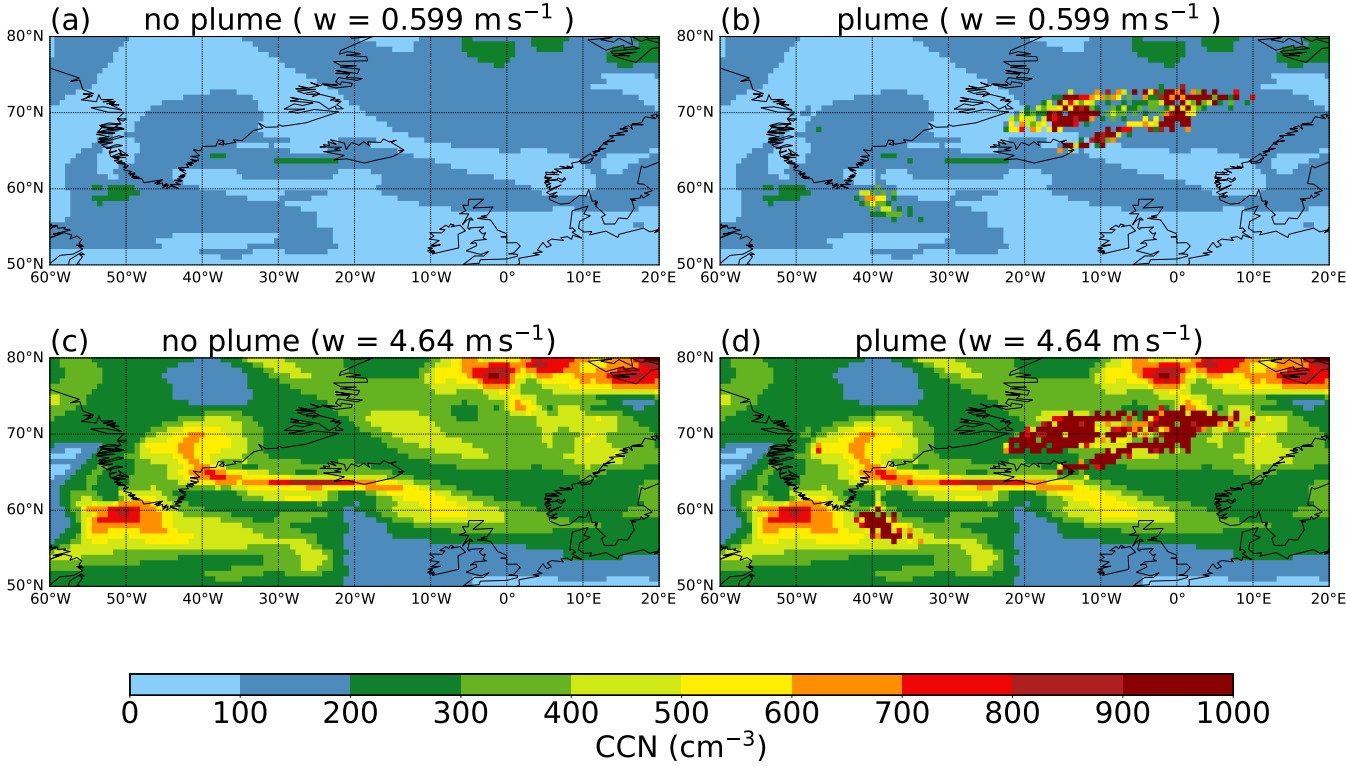

**Figure 3.** Number of column-mean activated CCN ($cm^{-3}$) for 2 September 2014 for two different vertical velocities (w = 0.55 m s$^{-1}$, upper panels, and w = 4.6 m s$^{-1}$, lower panels). Left panels (no-plume) correspond to background concentrations of aerosols and right panels (plume) correspond to scaled aerosol concentrations.

no-volcano simulations along with MODIS satellite retrievals for the 7 days starting on 1 September 2014. In the no-volcano simulation, there is no CCN enhancement due to the volcanic emissions (left column in Figure 3). Nevertheless, the grid points that are located inside of the volcano plume are compared to the ones outside the plume to assess differences due to different meteorological conditions.

$N_d$ is the first microphysical variable we assess. $N_d$ is not directly retrieved by the operational MODIS satellite retrievals. Instead, $r_e$ and $\tau_c$ are retrieved using the method described by Nakajima and King (1990). On the basis of such retrievals, assuming clouds that behave like adiabatic ones, $N_d$ can be computed as follows (Grosvenor et al., 2018):

$$N_d = \gamma \tau_c^{\frac{1}{2}} r_e^{-\frac{5}{2}}. \tag{1}$$

In this relation, $\gamma$ depends mainly on the adiabatic condensation rate and can be approximated as $1.37 \cdot 10^{-5}$ m$^{-\frac{1}{2}}$ (Quaas et al., 2006). In order to obtain $N_d$ by Equation 1 in our analyses both in simulations and MODIS, $r_e$ less than 4 $\mu$m and $\tau_c$ less than 4 were excluded from data set because they are less reliable (Nakajima and King, 1990). For consistency, $N_d$ is derived from the

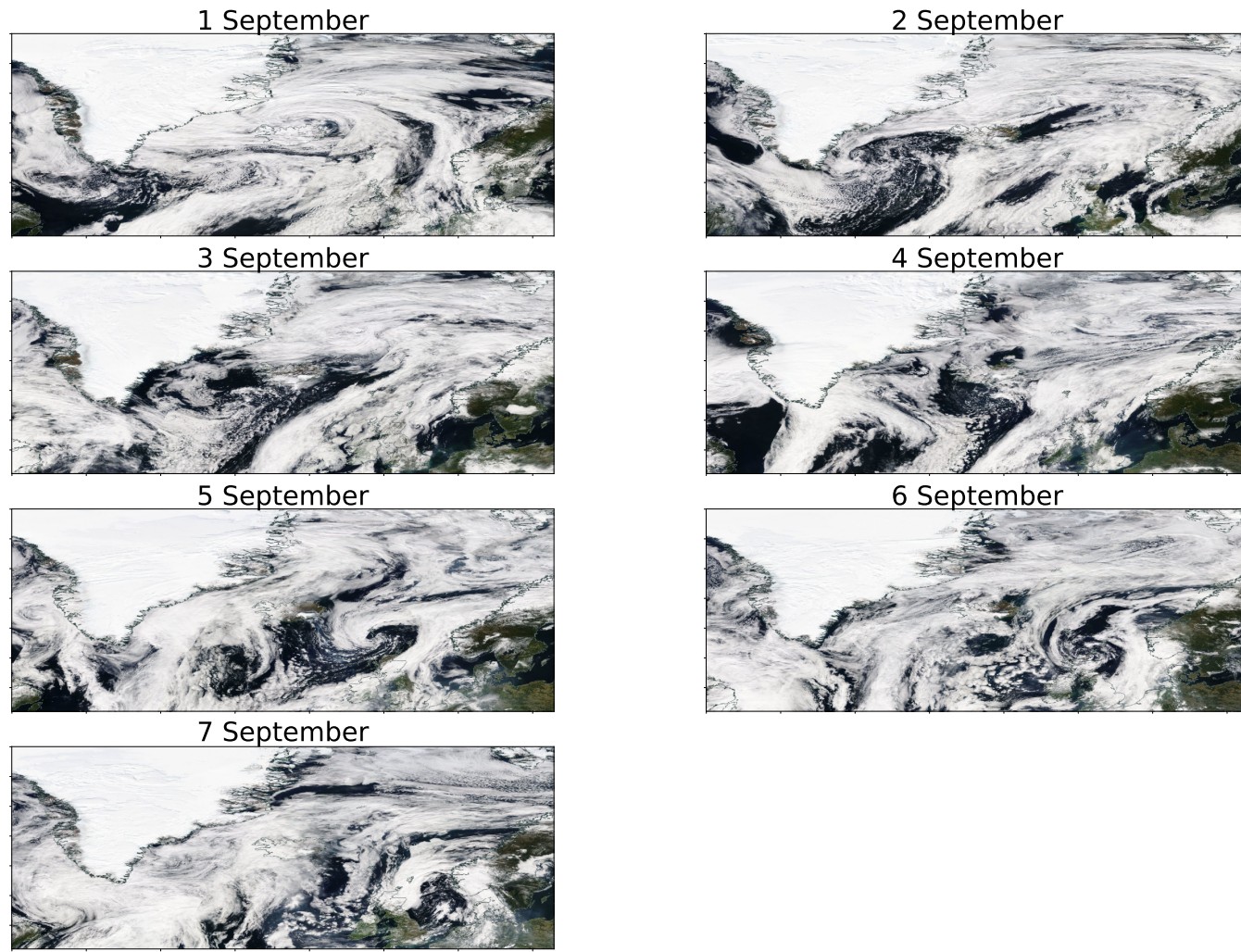

**Figure 4.** Visible image from MODIS-AQUA satellite retrievals from 1 to 7 September 2014. The satellite visible images were downloaded from per request by the following online ordering data systems: MODIS true-color images (https://worldview.earthdata.nasa.gov/)

COSP diagnostics of $\tau_c$ and $r_e$ (see section 2) in the same way as done in the MODIS retrievals. The model output is sampled at the time of the MODIS Aqua overpass of approximately 13.30 LST (Local Sidereal Time).

In the subsequent figures, in each panel the blue line is for the no-volcano run, the red line is for the volcano run and the black line is for the MODIS observations. Figure 5 shows the relative frequency distribution of $N_d$. In order to define the plume, marine pixels that correspond to the $SO_2$ concentrations in the lower troposphere exceeding 1 DU in Figure 2 were chosen. These are assumed to be located inside of volcano plume and the rest of the pixels are considered as outside of volcano plume. The right panel (outside of plume) indicates that the $N_d$ distribution outside of the volcano plume for both simulations are, as expected, very similar because the meteorology is the same and there is no additional aerosol. Comparing both simulations to

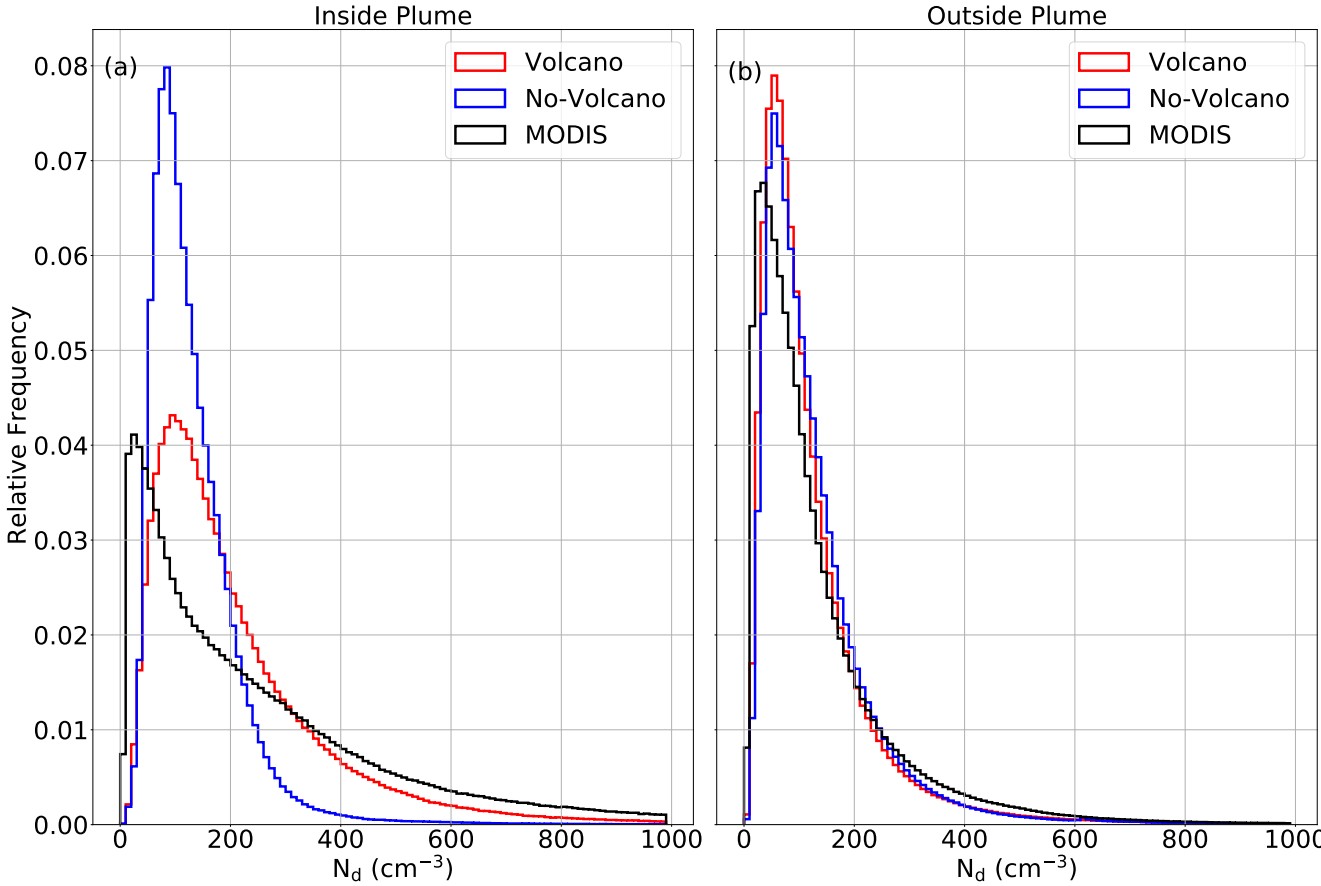

**Figure 5.** Relative frequency distribution of $N_d$ (cm$^{-3}$) for liquid clouds, inside of the plume (a) and outside the volcano plume (b) in the volcano simulation (red), the no-volcano simulation (blue) and MODIS Aqua level-2 data (black). The PDF shows the spatio-temporal variability for the MODIS overpass time for the seven days.

MODIS retrievals demonstrates that the simulated $N_d$ distribution is close to what is obtained from the satellite retrievals. In contrast, for the grid points inside the plume, it can be seen that $N_d$ is substantially enhanced in the volcano run compared to the no-volcano run as was expected due to the larger concentration of activated CCN inside the volcano plume. The $N_d$ distribution for MODIS shows that these observations are considerably closer to the volcano run with respect to the higher probability of large $N_d$ even if at lower concentrations there is a systematic discrepancy between MODIS data and both simulations. For such low concentrations, there is the possibility that the satellite data are biased (Grosvenor et al., 2018). For broken clouds, MODIS shows overly large $r_e$, which implies overly low $N_d$ (Eq. 1). Nevertheless, the results for the large $N_d$ concentrations, and the overall good agreement between the simulations and satellite retrievals (also outside the plume) allow for clear detection of the enhancement of $N_d$ inside the volcanic plume and its attribution to the volcanic aerosol.

| Variables | MODIS outside plume | MODIS plume en-hancement | no-vol outside plume | no-vol plume en-hancement | vol outside plume | vol plume enhance-ment |
|---|---|---|---|---|---|---|
| $N_{\mathrm{d}}$ (cm$^{-3}$) | 135 | 78% | 134 | 0% | 128 | 77% |
| LWP (g m$^{-2}$) | 149 | 7% | 151 | 6% | 151 | 30% |
| RWP (g m$^{-2}$) | - | - | 13 | 53% | 13 | 38% |
| Cloud fraction (%) | 52 | 29 % | 58 | 32% | 58 | 40% |
| $r_{\mathrm{e}}$ (μm) | 13 | -8% | 14 | 0% | 14 | -7% |
| $\tau_{\mathrm{c}}$ | 21 | 33% | 25 | -3% | 25 | 24% |
| All-sky Albedo | 0.39 | 18% | 0.33 | 27% | 0.35 | 42% |
| Cloudy-sky Albedo | 0.44 | 9% | 0.46 | 0% | 0.45 | 7% |

**Table 1.** Mean values for $N_{\mathrm{d}}$, LWP, RWP, total cloud fraction and albedo at top of atmosphere for MODIS (CERES for the albedo), the no-volcano simulation and volcano simulation. The values are computed for outside of plume and enhancement inside of plume which computed as $\left(\frac{\text{mean for inside of plume} - \text{mean for outside of plume}}{\text{mean for outside of plume}}\right)$.

The mean values for $N_{\mathrm{d}}$ are listed in Table 1. The mean $N_{\mathrm{d}}$ in the plume, compared to the mean of the distribution outside the plume, is enhanced by 77 % in volcano run compared to no (0 %) change in the no-volcano run. The enhancement value in MODIS is 78 % which almost exactly is the same as in the volcano run. The mean $N_{\mathrm{d}}$ outside the plume is 134 cm$^{-3}$, 128 cm$^{-3}$ and 135 cm$^{-3}$ for no-volcano, volcano simulations and MODIS respectively, showing that outside of plume $N_{\mathrm{d}}$ didn't change considerably between simulations because the meteorology is same and there is no additional activated CCN, and showing good consistency between both model runs and the satellite retrievals. Table 1 further lists the mean values and changes for $r_{\mathrm{e}}$ and $\tau_{\mathrm{c}}$. The effective radius decreased inside of plume by 7 % compare to outside the plume in volcano simulation. In the no-volcano simulation, there is no difference in $r_{\mathrm{e}}$ inside vs. outside the plume. In the MODIS retrievals, $r_{\mathrm{e}}$ decreased by 8 % inside the plume compared to outside the plume, very similar to the change in the simulation. This is consistent with the agreement in plume enhancement for $N_{\mathrm{d}}$. Also the cloud optical thickness shows a consistent increase in MODIS as in the volcano simulation, whereas the no-volcano simulation shows a (very slight) decrease in $\tau_{\mathrm{c}}$ inside the plume.

Figure 6 shows the same analyses as Figure 5 but for LWP. The distribution of LWP for the region outside the volcano plume is not significantly different between the two simulations, as expected. The mean values for LWP (Table 1) in both simulations

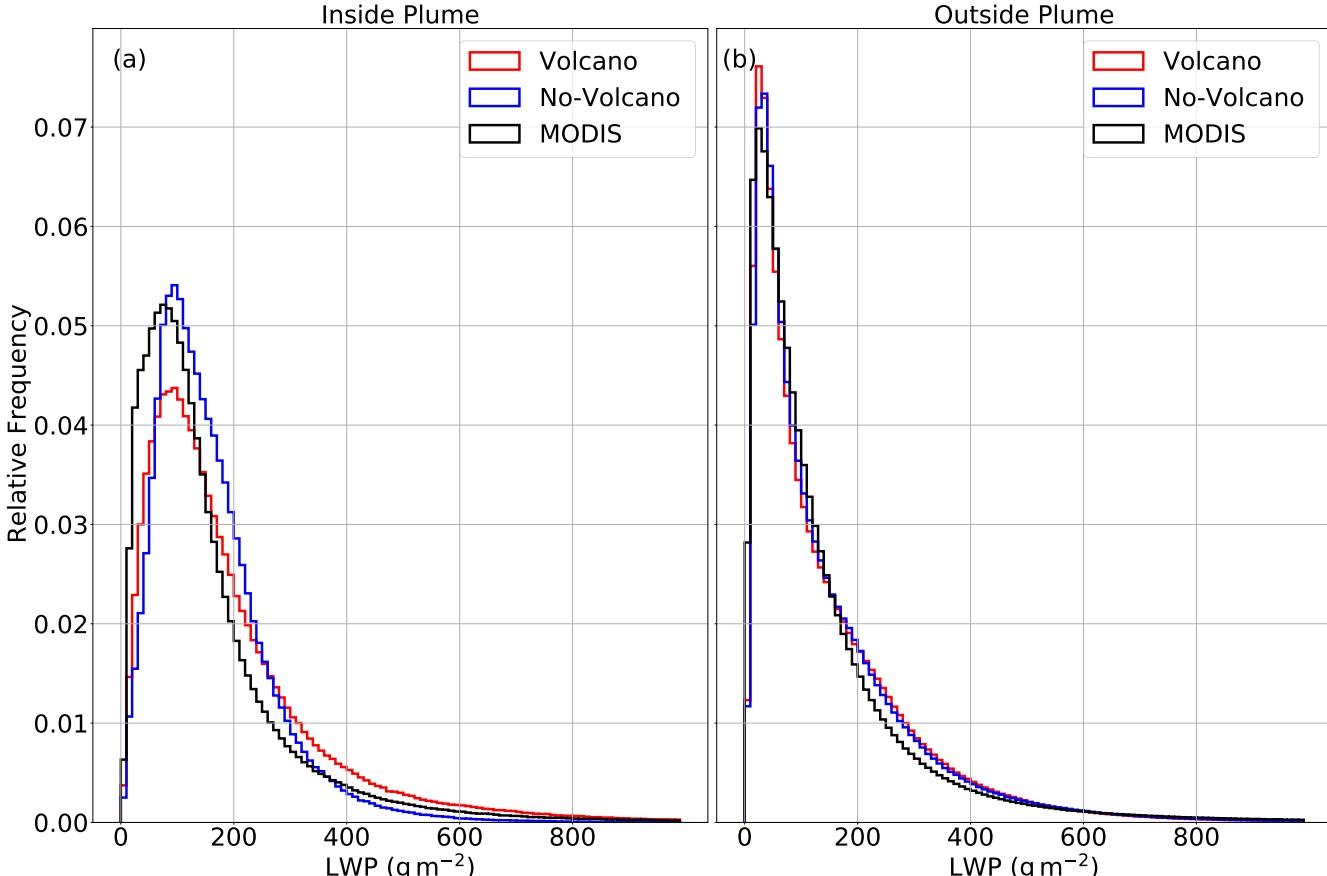

**Figure 6.** As Figure 5, but for LWP (g m$^{-2}$).

is the same at 151 g m$^{-2}$; furthermore, the MODIS mean value of 149 g m$^{-2}$ is close to the simulations which demonstrate the accuracy of clouds simulations. This is also true for the entire distribution (Figure 6). Considering the simulated profiles, in the simulation with volcano emissions included, there is a decrease in the probability of clouds with lower LWP and an increase

in the probability of clouds with higher LWP compared to the no-volcano simulation. The MODIS distribution for LWP inside the plume indicates that the probability for clouds with lower LWP is less than what the simulations show, but the probability for clouds with higher LWP is more than in the no-volcano run, albeit also less than in the volcano run. In terms of the mean values for LWP (Table 1) for inside of plume, the simulations indicate a slight enhancement (+6 %) attributable to the different weather conditions (plume enhancement in the no-volcano run), and a strong enhancement (+30 %) in the volcano run. The

difference suggests that the model shows an LWP enhanced by 24% due to additional CCN inside of volcano plume. MODIS, however, is very close to the result of the no-volcano run for the average values. This almost zero enhancement on average, however, seems to come about by a decrease in LWP for the clouds with low LWP, and an enhancement of LWP for large LWP

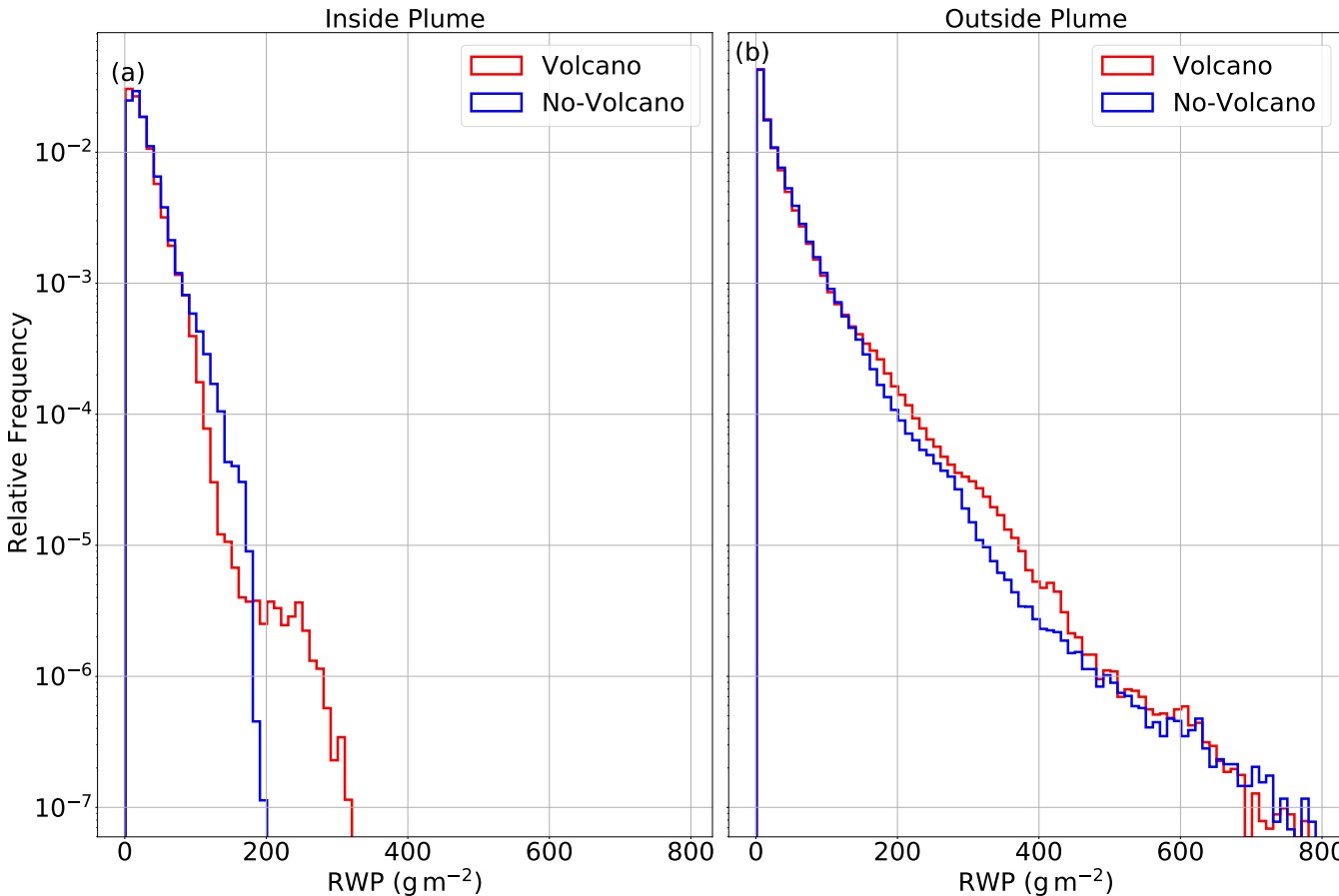

**Figure 7.** Relative frequency distribution of RWP profile in logarithmic scale for inside of plume (a) and outside of volcano plume (b) in volcano simulation (red) and no-volcano simulation (blue).

values (Figure 6). This is qualitatively consistent with the results of the ICON-NWP model. The model, however, exaggerates the increase in large LWP values, leading to the exaggerated mean increase.

The question is now what is the underlying process leading to an increase in LWP in the volcano simulation? One reason is the suppression of precipitation (e.g., Seifert et al., 2012). Therefore, the distribution of rain water path (RWP) was analyzed to investigate the alteration of precipitation inside and outside the volcano plume in both, the volcano and no-volcano simulations. The comparison is shown in Figure 7. Since the precipitation information is not available from MODIS or other satellite retrievals, RWP is only depicted for the simulations. Inside the volcano plume, there is a decrease in light rain and an increase in

heavy rain for the volcano simulation, compared to the no-volcano simulation. In terms of mean values for RWP (Table 1), there is a decrease in the volcano run by 15 % on average, while the precipitation profile for outside of plume is quite similar which is in the agreement of the fact that LWP for outside of plume didn't alter significantly. In the volcano simulation compared to the no-volcano simulation, for the region inside the plume, RWP of less than about $180\,\mathrm{g\,m^{-2}}$ occur less frequently. This is

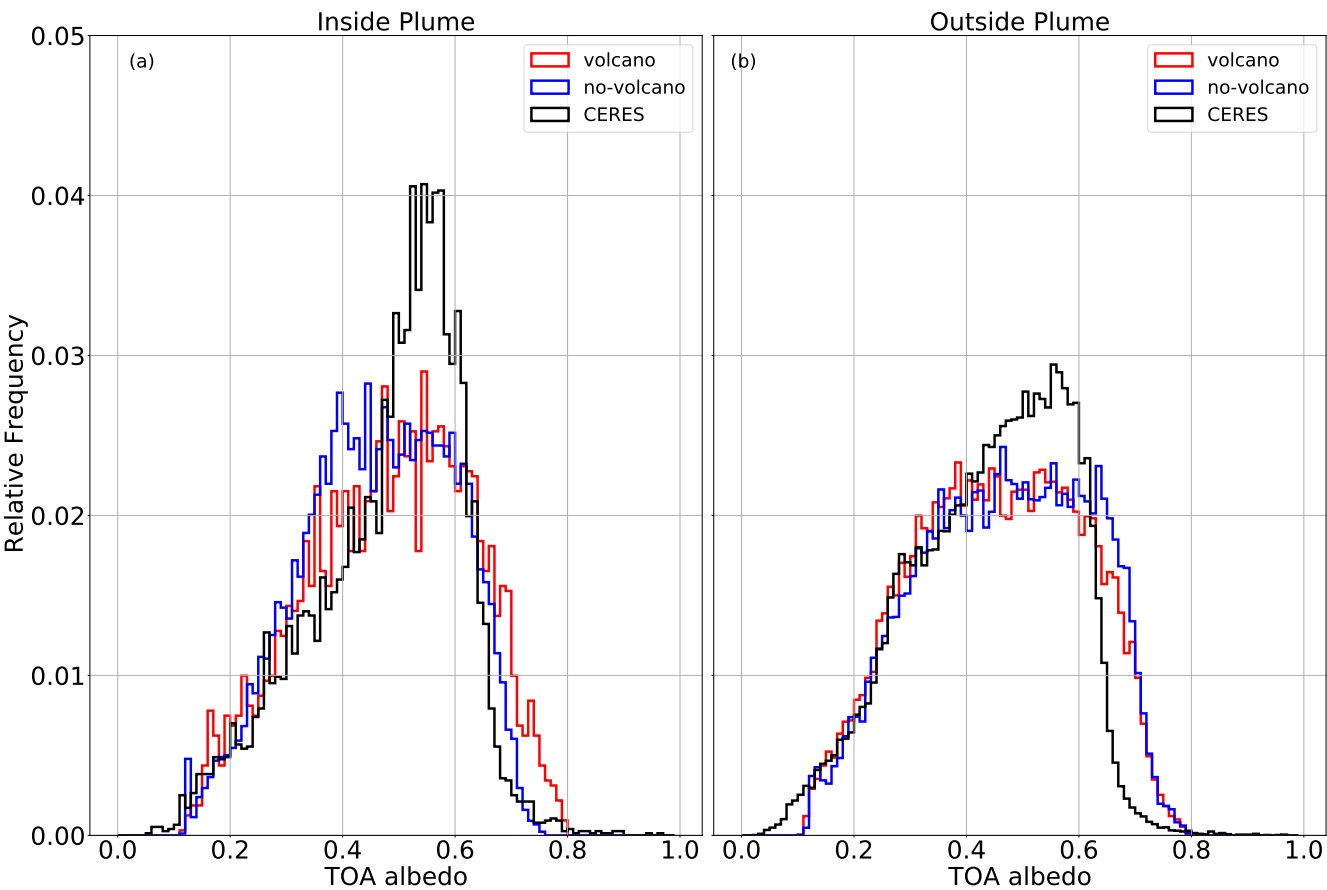

**Figure 8.** Relative frequency distribution of TOA albedo profile for inside of plume (a) and outside of volcano plume (b), in volcano simulation (red), no-volcano simulation (blue) ,and CERES level-2 footprint data (black).

because at larger $N_\mathrm{d}$, thus smaller droplets, the occurrence of light rain is suppressed. Cloud droplets must grow larger, leading
to deeper clouds in order to reach the size that they can start to precipitate. This leads to a shift in the LWP distribution to the higher values inside the plume. In turn, these deeper clouds in which drops have more water available for growth produce heavier precipitation. Therefore there is an enhancement in the probability of heavy rain (RWP larger than about $180\,\mathrm{g\,m^{-2}}$), compared to no-volcano simulation inside of volcano plume, even if the average RWP (Table 1) does not change significantly. Moreover, suppression in precipitation can also lead to enhancement in cloud horizontal extent (cloud fraction). Therefore, the
modification in cloud fraction was examined in simulations and MODIS. The analyses for mean values of total cloud fraction in Table 1 demonstrates that, in the volcano simulation, cloud fraction is enhanced in the plume compared to outside the plume by 40 %, while the enhancement is only 32 % in the no-volcano simulation. However, even in the no-volcano simulation, cloud fraction inside of plume is higher than outside of plume by 32 % due to the different weather conditions, and this is consistent with what MODIS shows (29 %).

## 4 Implications for the radiative impact

Finally, the effect on radiation (indicative of the effective radiative forcing due to the modification of cloud properties by the volcanic aerosol) is examined. Therefore the TOA albedo was analyzed inside and outside of plume in simulations and CERES level-2 footprint data (Su et al., 2015). For the comparison, the simulation output was remapped by distance weighted average remapping of the four nearest neighbor values method to 20 km horizontal resolution to be consistent with the resolution of the CERES footprint. In Figure 8 TOA albedo for the cloudy sky is depicted for inside and outside the volcano plume for both simulations and CERES data. Grid points with $SO_2$ concentrations in the lower troposphere exceeding 1 DU are considered to constitute the plume, and $SO_2$ concentration was obtained from OMPS satellite retrievals which are in 50 km footprint data in level-2. We remapped the level-2 data into the 50 km resolution, and due to the fact that CERES products in 20 km resolution, it has sufficient resolution to identify the plume. Clear sky was excluded because, in the model, no aerosol-radiation interactions are considered, but in the CERES this effect is in the data and would bias the analysis for clear sky. An additional important aspect that should be considered, is that the TOA albedo distribution is considered here for liquid clouds with $\tau_c$ more than 4 because in obtaining $N_d$ the data with $\tau_c$ less than 4 were excluded as well. Considering the TOA albedo distribution inside the plume, it is seen that in the volcano simulation, there is a higher probability for TOA albedo larger than 0.6 compared to the no-volcano simulation. In the CERES data, there is a peak at TOA albedo between 0.4 and 0.6 that is not as pronounced in either simulation. In turn, the probability for TOA albedo larger than 0.7 is smaller in the data than in both simulations. This bias, however, is clear outside the plume but much less so inside the plume - possibly indicative of the albedo enhancement due to the volcanic aerosol.

For the mean values (Table 1), in turn, clear sky data were taken into account to be able to see the influence of cloud fraction changes on modifying TOA albedo. The difference in mean values between inside and outside the plume in the volcano simulation is 15 % larger compared to no-volcano simulation. In CERES data there is an 18 % enhancement inside the volcano plume compared to outside the plume. When compared to the difference between inside and outside the plume in the no-volcano simulation (27 %), it is difficult to conclude that there is a signal of alteration in TOA albedo in CERES data. We also analyzed cloudy sky TOA albedo mean values in simulations and CERES. The values in Table 1 demonstrate an enhancement of 9 % in CERES and 7 % in volcano simulation while no changes were obtained in no-volcano simulation. The daily mean incoming solar radiation was obtained 260 W m$^{-2}$; therefore, effective radiative forcing except cloud cover effect can be estimated as 10 W m$^{-2}$ in CERES and 8  W m$^{-2}$ in volcano simulation.

## 5 Conclusions

In this study, the impact of aerosols emitted by the Holuhraun volcanic eruption on liquid clouds was assessed. For this, we used a pair of cloud-system resolving simulations with and without the enhancement in CCN due to the volcanic emission, as well as MODIS and CERES satellite retrievals. The COSP simulator was implemented in the model to allow for an apples-to-apples comparison between the simulations and satellite data. To identify the impact of the additional aerosol on cloud microphysical properties, areas located inside and outside the volcano plume were compared in terms of their statistical distributions. In the

no-volcano simulation, only the differences in weather conditions are sampled. In the in volcano simulation, in addition, there is the effect of the CCN enhancement on the clouds. To the extent the inside vs. outside-plume difference is consistent between the satellite retrievals and the volcano simulation, but not between the satellite retrievals and the no-volcano simulation, detection and attribution of the effect of the aerosol on the clouds is achieved. Our analyses indicated that $N_d$ concentration is clearly enhanced inside the volcano plume. This enhancement by almost 80 % is attributable to the additional CCN inside the volcano plume. Our scientific goal in this study was to examine how LWP and cloud fraction respond to the enhancement of the $N_d$ in the volcanic plume. The analysis reveals that in the simulations and MODIS, the LWP is increased inside the plume compared to outside the plume. However, the mean increase in MODIS is very close to the result of the no-volcano run. This almost zero enhancement in MODIS on average is because of decrease in LWP for the clouds with low LWP, and an enhancement of LWP for large LWP values which is consistent with the results of the ICON-NWP model, nevertheless, the model, exaggerates the increase in large LWP values. In the model the reason for the enhancement of LWP in the volcano simulation was the decrease in precipitation compared to no-volcano simulation by 15 % on average, due to a decrease in light rain in the volcano simulation compared to the no-volcano simulation. When light rain is depressed, clouds droplets must grow deeper in order to reach the droplet size at which precipitation is initiated. This leads to a shift in the LWP distribution to the higher values. Since cloud droplets grow deeper, they precipitate heavier because they have to fall through more clouds droplets. Examining cloud fraction (only possible for the mean value) demonstrates that the cloud fraction also increased inside the plume in the volcano simulation compared to the no-volcano simulation. Similar to the result for LWP, this mean increase cannot be attributed confidently to the volcanic aerosol. It is unclear for the MODIS data, how much change in cloud fraction between inside and outside the plume is due to the enhancement of cloud lifetime due to the additional CCN and how much simply is because of different weather. To learn about the climate implications, it is essential to identify how the planetary albedo differs inside and outside the volcano plume. In this study, the difference in increase of TOA albedo between inside and outside the volcano plume in the volcano and no-volcano simulations was quantified at 42% when considering the volcanic aerosol vs. only 27% without it, however it is, again, not possible to attribute the enhancement in TOA albedo in the CERES observations.

Overall, the results from this detailed analysis using level-2 satellite observations and cloud-system resolving simulations confirm the key result of Malavelle et al. (2017) that there is a clear, detectable and attributable impact of the volcanic aerosol on $N_d$, but there is on average only a very small, not attributable, effect on both LWP and cloud fraction. This net result for the case of the Holuhraun volcano for LWP comes about by a slight enhancement of LWP for large-LWP clouds compensated for by a decrease in LWP in low-LWP clouds.

*Data availability.* The ICON model outputs are stored at the German climate computing center (DKRZ) and are available upon request to the corresponding author. The MODIS data were downloaded from the Atmosphere Archive  Distribution System (LAADS) Distributed Active Archive Center (DAAC), located in the Goddard Space Flight Center in Greenbelt, Maryland (https://ladsweb.nascom.nasa.gov). CAMS re-analyses are available from the Atmosphere Data Store (ADS), either interactively through its download web form or by using the CDS API service (https://confluence.ecmwf.int/display/COPSRV/Copernicus+Atmosphere+Monitoring+Service+-+CAMS). OMPS data was down-

## Appendix A: Look-up table of potentially activated CCN number concentrations

The look-up table consists of potentially activated CCN number concentrations for 10 specific vertical velocities and height for
each hybrid-sigma-pressure level (60 levels) and 3 hour interval. This look-up table was calculated offline for 1 to 7 September 2014 (the period of simulation). In order to show the range of values, we choose 2 September and compute its daily mean. The model level corresponding to approximately (850 hPa) was chosen. The table A1 summarizes each specific vertical velocity that has been used in the box model for computations of potentially activated CCN concentration. The value range is shown as the mean value for the whole domain and the first and the third quartile of grid point values.

| 2 September 2014 | | | |
|---|---|---|---|
| Variables | First Quartile ($cm^{-3}$) | Mean Value ($cm^{-3}$) | Third Quartile ($cm^{-3}$) |
| CCN-act ($w = 0.01$ m s$^{-1}$) | 5 | 7 | 8 |
| CCN-act ($w = 0.0278$ m s$^{-1}$) | 20 | 26 | 32 |
| CCN-act ($w = 0.0774$ m s$^{-1}$) | 54 | 73 | 87 |
| CCN-act ($w = 0.215$ m s$^{-1}$) | 117 | 166 | 204 |
| CCN-act ($w = 0.599$ m s$^{-1}$) | 230 | 334 | 414 |
| CCN-act ($w = 1.67$ m s$^{-1}$) | 406 | 605 | 753 |
| CCN-act ($w = 4.64$ m s$^{-1}$) | 639 | 994 | 1235 |
| CCN-act ($w = 12.9$ m s$^{-1}$) | 918 | 1492 | 1842 |
| CCN-act ($w = 35.9$ m s$^{-1}$) | 1219 | 2070 | 2534 |
| CCN-act ($w = 100$ m s$^{-1}$) | 1528 | 2691 | 3271 |

**Table A1.** Look-up table of potentially activated CCN.

## A1 Vertical profile of activated CCN

The scaling of CCN was done by computing distribution of scaling based on the enhancement of SO2 inside the plume relative to the mean SO2 value outside of the plume in the lower troposphere (up to 3 km). Then the sulfate concentrations in CAMS reanalyses inside of the plume were scaled at each level by the computed ratio. So the sulfate aerosol concentration at each
365 level was scaled with the same ratio but the concentration of the sulfate is not the same at each level because the background concentration is different at each level. In the next step, the box model was employed on the scaled sulfate aerosol concentration, and the scaled CCN profile was obtained. To determine the vertical distribution of CCN more specifically, figure A1 shows the

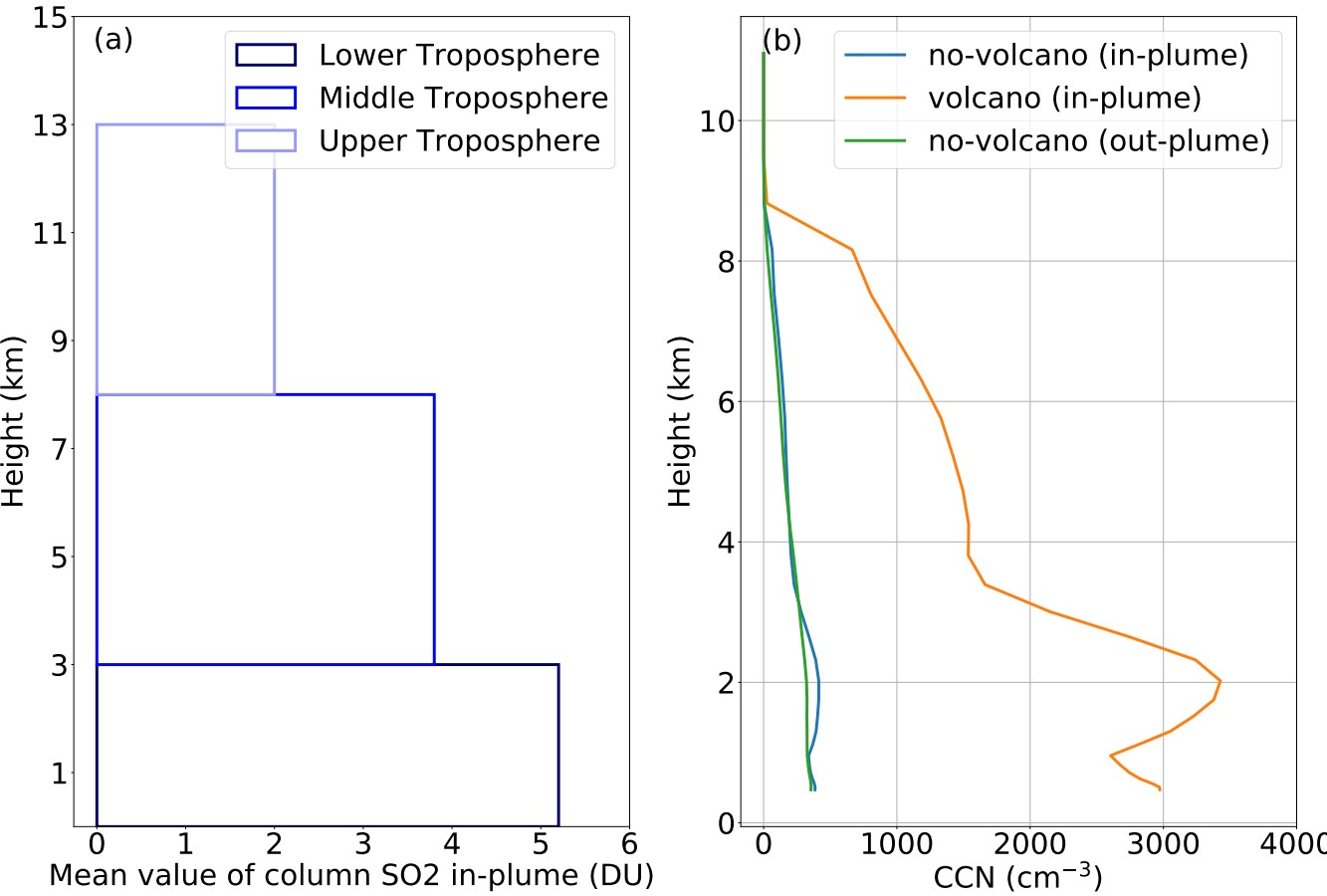

**Figure A1.** Mean value of column SO2 concentration inside of plume in the lower, middle, and upper troposphere in OMPS retrievals (panel a) along with the vertical profile of mean CCN concentration inside of the plume in the no-volcano and volcano run and outside of the plume in the no-volcano run (panel b) for just one specific amount of vertical velocity (0.559 m/s) on 2 September 2014.

mean value of column SO2 concentration inside of the plume in the lower, middle, and upper troposphere in OPMS retrievals along with the vertical profile of mean CCN concentration inside of the plume in the no-volcano and volcano run and outside of the plume in the no-volcano run for one specific vertical velocity (0.559 m/s) on 2 September 2014.

*Author contributions.* MH and JQ conducted this study. JK helped with setting up and running ICON-NWP. KB contributed to producing data for the study. MH prepared the model and observational data. All authors contributed to the interpretation of the results. MH produced the manuscript with the aid of all co-authors.

*Competing interests.* The authors declare that they have no conflict of interest.

*Acknowledgements.* We appreciatively acknowledge funding by the German Research Foundation (Deutsche Forschungsgemeinschaft, DFG) for funding via the VolCloud project (GZ QU 31123-1) as part of the VolImpact research unit. JQ further acknowledges funding by the European Union via its Horizon 2020 project CONSTRAIN (GA 820829). We are thankful for the valuable collaboration with colleagues in the research unit VolImact, especially within VolCloud with the Karlsruhe Institute of Technology, Corinna Hoose and Fatemeh Zarei. We are thankful to the German Meteorological Service (DWD) and the Max Planck Institute for Meteorology to provide the ICON model to the 380   research community, and German Climate Computing Center (Deutsches Klimarechenzentru, DKRZ) to provide the resources to conduct the simulations. We also thank NASA for providing the satellite retrievals employed in this study. We also thank (https://worldview.earthdata. nasa.gov/) website to provide satellite true images. We further acknowledge funding by the Open Access Publishing Fund of Leipzig University supported by the German Research Foundation within the program Open Access Publication Funding.

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
