# Peer review of "Impact of Holuhraun volcano aerosols on clouds in cloud-system resolving simulations"

_Atmospheric Chemistry and Physics, 2022_

## Referee Comment (RC2)

**Review of "Impact of Holuhraun volcano aerosols on clouds in cloud-system resolving simulations" by Haghighatnasab et al., submitted to Atmospheric Chemistry and Physics (ACP)**

[Article#: acp-2022-38]

This report contains overall, major and minor comments from this reviewer to the manuscript.

**A summary of the manuscript and overall assessment:**

Recommendation: Major revision

This study performed sensitivity experiments for a region around the North Atlantic to investigate the effects of volcanic smoke aerosols as cloud condensation nuclei (CCN) associated to the eruption in Holuhraun on the cloud properties in the volcanic smoke trails. A scientific goal is to investigate how liquid water path (LWP) and cloud fraction change in response to increase in CCN and subsequently cloud droplet number (Nd) in the volcanic plumes. Regional cloud-resolving simulations with approximately 2.5 km grid spacings were conducted for a week when emission of SO2 from the eruption was clearly identified in satellite observations. The simulation results were compared to satellite observations for cloud to check how the observed difference in cloud properties between in and outside the volcanic plumes was reproduced in the simulations. The simulation considering effects of volcanic smokes replicated the observed increase in Nd in the plumes. However, the same simulation overpredicted increase in significantly LWP and slightly in cloud fraction.

I think the current investigation and discussion on the relationship between LWP and Nd or Na (aerosol or CCN number concentration) is insufficient. As described in the current manuscript and in reports from model intercomparison projects (e.g., Quaas et al. 2009), conventional global aerosol transport models tend to overpredict increase in LWP in response to increase in Na or Nd, compared to global-scale satellite observations. However, several recent modeling studies, particularly using high-resolution (cloud- or large-eddy-resolving-scale) models at a regional or global scale, reported little change or even decrease in LWP in the response, according to condition. Their results may be more consistent with the finding in Malavelle et al. (2017), which is the case in this study. First, the manuscript should include careful literature reviews about the advances in recent modeling studies on the sensitivity of LWP to variation in aerosol, CCN, or cloud drop number concentration. Then, more discussion and investigation are needed to examine

why the results of the cloud-resolving model simulation in this study contradict findings in some of those recent modeling studies as well as the observation results for the volcanic smoke case. I think at least this effort has to be done toward being acceptable. I have several other major comments listed in the following section. The authors are encouraged to revise the manuscript to improve the quality and readability.

**Major comments:**

1. LWP-Nd

As an example of limited-area large-eddy simulation for aerosol-cloud interaction, Seifert et al. (2015) conducted an extensive series of sensitivity simulations. They reported a negative lifetime effect (unchanged LWP and decrease in cloud cover with increasing Nd) in addition to positive one which has been seen in other previous LES studies, depending on the meteorological condition and the stage of cloud life cycle. Similar dependency of the sensitivity on meteorological condition and cloud regime was found in other LES studies (e.g., Lebo and Feingold 2014). On the other hand, Sato et al. (2018) conducted one-year global cloud-resolving simulation to examine the sensitivity. They successfully reproduced negative $\lambda c$ (the definition can be found in the paper) seen in satellite observations, mostly over regions where cumulus was dominant. They suggested that evaporation process of cloud droplets around cloud top was important to resulting in negative values. More details of the discussion can be found in the paper. As I wrote in the overall comment, since some of other modeling studies could reproduce negative sensitivity, the authors should make efforts to examine and explain why the current simulation could not do it in discussion together with findings in previous studies not limited to those shown above. I understand models have various uncertainty and hence often cannot reproduce observations. But the manuscript should show some advances toward the next step.

Seifert, A., Heus, T., Pincus, R., & Stevens, B. (2015). Large-eddy simulation of the transient and near-equilibrium behavior of precipitating shallow convection. Journal of Advances in Modeling Earth Systems, 7(4), 1918-1937.

Lebo, Z. J., & Feingold, G. (2014). On the relationship between responses in cloud water and precipitation to changes in aerosol. Atmospheric Chemistry and Physics, 14(21), 11817-11831.

Sato, Y., Goto, D., Michibata, T., Suzuki, K., Takemura, T., Tomita, H., & Nakajima, T. (2018). Aerosol effects on cloud water amounts were successfully simulated by a global cloud-system resolving model. Nature Communications, 9(1), 1-7.

2. Meteorological and cloud information of the target case

The manuscript should show what meteorological condition and what types of cloud were dominant in the period and the domain for the simulations. These information is quite important in the discussion because previous studies, e.g., in comment #1, showed some dependency of the aerosol-cloud interaction on those factors. Some MODIS true-color images may help it. And another question, is only warm-topped cloud with cloud top temperature over 273.15 K analyzed and is the other cold-topped cloud excluded?

3. Vertical distribution of the volcanic aerosol plume

The OMPS satellite retrieval products were used to identify the column total SO2, and then sulfate aerosol mass mixing ratio was calculated based on the difference in column total SO2 between in and outside the volcanic (around Ln. 143). But I think the vertical profiles of SO2 and sulfate aerosol concentrations might differ between, because they might be contaminated in limited vertical layers into which smoke was injected. How did the authors consider the vertical injection or vertical distribution of the volcanic aerosol plume? Or, maybe I am confused, does the model not need the information of vertical distribution of aerosol but just use column-integrated value to calculate activated CCN concentration at each vertical level?

4. Definitions of LWP in MODIS product and the simulation

It is clearly written that Nd in the simulations were calculated using a satellite simulator through same pathway as for the MODIS products. But what about LWP? The definition of LWP has large uncertainty between the satellite products and the model simulation even using a simulator because bulk cloud microphysics has a category gap between cloud water and rain. This is problem in the radiative transfer calculation in simulator to determine LWP that is consistent with that in satellite products. This problem may affect the calculation of other variables such as Nd also.

5. Discussion on cloud fraction

I think 2.5 km model grid spacing may be still coarse for comparative discussion of cloud fraction over ocean with the Level-2 MODIS-Aqua cloud product (swath 1km). The model simulation might miss parts of scattered shallow cumulus over ocean and overemphasize extent of deeper cloud. This might contribute the overprediction of positive cloud lifetime effects on cloud fraction in the plumes in the simulation too. The shallow convection parameterization of Tiedtke (1989) has no effects on the calculation of the cloud fraction, correct?

6. CERES 20 km resolution

Is the 20 km resolution of the CERES products enough to distinguish in and outside the smoke plume? The spatial scales of the smoke trails are unclear to me. And what algorithm was used for remapping the model results from the native model grid structure to those with 20 km grid spacing? The selection of the algorithm may strongly affect the results because it was from fine to very coarse grid structures.

**Minor comments:**

Ln. 39: "cloud" => "could"

Ln. 84: Same question as in major comment #6, what algorithm was used for remapping?

Around Ln. 110: Can you summarize the variables in the look-up table and the value ranges into a table?

Table 1: Could you add comparison of $\tau c$ and re into Table 1 too?

Figs. 2 and 3: Please add lines of latitude and longitude to the maps.

Ln. 253-255: These sentences are a bit awkward. Please rephase and improve the readability.

Ln. 256-257: The sentence is confusing. The vertical axis of the plots in Fig. 6 is at a log-scale. The frequency of high RWP over 200 gm-2 in the volcano simulation is quite or neglectably small, and the difference in mean RWP in the plumes is due to the difference in the frequency of lower RWP values.

---

## Author Response (AR1)

**Response to the Reviewer's comments**

*We would like to thank the reviewers for the effort in helping us improve the manuscript. Below we respond point-by-point to the comments, with the reviewer comments in black, our responses in black and the changes in revised manuscript in blue. The line numbers are for the revised manuscript version.*

**Response to referee comments #1**

*In this study, the authors conducted numerical simulations targeting the eruption of Holuharaun volcano by the NWP mode of ICON, and investigated the impacts of aerosol emitted from the volcano on $N_d$, LWP, cloud fraction, and cloud albedo. Through their analyses the authors clarified that the impact of the aerosols on the LWP and cloud fraction was mainly explained by the difference of the meteorological condition between inside and outside plume, although the increase of $N_d$ was clearly seen as the impact of the aerosols. In my understanding, the response of the LWP and cloud fraction to the aerosol variation are featured topics in the scientific community and the results of this study are interesting. So, I encourage the authors to conduct this study. Most part of the manuscript is well written, but there are several issues to be addressed. Based on the descriptions outlined above, my decision is "major revision", and I encourage the authors to revise the manuscript.*

We thank the reviewer for their assessment of our manuscript. The review helped to improve the manuscript significantly.

**Major comments**

*1. The authors discussed the effects of the emitted aerosol on the radiative forcing based on the results of cloud albedo. However, the goal of this study is to understand how LWP and cloud fraction respond to the aerosol variation, as the authors indicate in the body of the manuscript. So, the discussion about the radiative forcing and cloud albedo will make readers to confuse the main topic of this study. Of course, I understand that the radiative forcing and cloud albedo are really important for the climate study, but focusing on the LWP and cloud fraction makes the main message of this study clear. The discussion about the cloud albedo and radiative forcing in discussion section, which can be created in the revised manuscript is better, or the discussion of them in supplementary information is another option.*

We appreciate the reviewer's suggestion. A new section "Implications for the radiative impact" is now added in the revised manuscript before the "conclusion" section and all the information regarding

radiative forcing is moved to this section to tone down this aspect of the analysis, in light of the reviewer's concern.

*Lines (284-310):*

**Section 4 : Implications for the radiative impact**

Finally, the effect on radiation (indicative of the effective radiative forcing due to the modification of cloud properties by the volcanic aerosol) is examined. Therefore the TOA albedo was analyzed inside and outside of plume in simulations and CERES level-2 footprint data (Su et al., 2015). For the comparison, the simulation output was remapped by distance weighted average remapping of the four nearest neighbor values method to 20 km horizontal resolution to be consistent with the resolution of the CERES footprint. In Figure 8 TOA albedo for the cloudy sky is depicted for inside and outside the volcano plume for both simulations and CERES data. Grid points with $SO_2$ concentrations in the lower troposphere exceeding 1 DU are considered to constitute the plume, and $SO_2$ concentration was obtained from OMPS satellite retrievals which are in 50 km footprint data in level-2. We remapped the level-2 data into the 50 km resolution, and due to the fact that CERES products in 20 km resolution, it has sufficient resolution to identify the plume. Clear sky was excluded because, in the model, no aerosol-radiation interactions are considered, but in the CERES this effect is in the data and would bias the analysis for clear sky. An additional important aspect that should be considered, is that the TOA albedo distribution is considered here for liquid clouds with $\tau_c$ more than 4 because in obtaining $N_d$ the data with $\tau_c$ less than 4 were excluded as well. Considering the TOA albedo distribution inside the plume, it is seen that in the volcano simulation, there is a higher probability for TOA albedo larger than 0.6 compared to the no-volcano simulation. In the CERES data, there is a peak at TOA albedo between 0.4 and 0.6 that is not as pronounced in either simulation. In turn, the probability for TOA albedo larger than 0.7 is smaller in the data than in both simulations. This bias, however, is clear outside the plume but much less so inside the plume - possibly indicative of the albedo enhancement due to the volcanic aerosol. For the mean values (Table 1), in turn, clear sky data were taken into account to be able to see the influence of cloud fraction changes on modifying TOA albedo. The difference in mean values between inside and outside the plume in the volcano simulation is 15 % larger compared to no-volcano simulation. In CERES data there is an 18 % enhancement inside the volcano plume compared to outside the plume. When compared to the difference between inside and outside the plume in the no-volcano simulation (27 %), it is difficult to conclude that there is a signal of alteration in TOA albedo in CERES data. We also analyzed cloudy sky TOA albedo mean values in simulations and CERES. The values in Table 1 demonstrate an enhancement of 9 % in CERES and 7 % in volcano simulation while no changes were obtained in no-volcano simulation. The daily mean incoming solar radiation was obtained 260 W m$^{-2}$ ; therefore, effective radiative forcing except cloud cover effect can be estimated as 10 W m$^{-2}$ in CERES and 8 W m$^{-2}$ in volcano simulation.

*2. In the section 3, the authors suggest that the decrease of the light rain and increase of the heavy rain in volcano simulation based on Fig. 6a. In addition, the authors suggest that the enhancement of the RWP is decreased by 15 % and no difference in RWP over outside of plume based on Table 1. I agree these suggestions, but it is not clear about why the decrease of the light rain and the increase of the heavy rain can result in the decrease of the enhancement of RWP and as consequence, increase of LWP. The manuscript is not so long at current version, and therefore, the author can add detailed descriptions about the reason. Such descriptions will help readers to understand the authors' suggestions more clearly.*

We appreciate the reviewer's suggestion. We conclude in the manuscript that the decrease of light rain and the increase of heavy rain results in the enhancement of LWP inside of the volcano plume. The reason for this conclusion is that when light rain is depressed, cloud droplets must grow larger in order to reach the size that they can start to precipitate which leads to a shift in the LWP distribution through the higher values, and if cloud droplets grow larger, they produce heavier precipitation. Therefore there is an enhancement in the probability of heavy rain compared to no-volcano simulation inside of volcano plume, even if the average RWP (Table 1) does not change much. These informations are added in the revised manuscript.

 *Lines (271-277):*

In the volcano simulation compared to the no-volcano simulation, for the region inside the plume, RWP of less than about 180 g m$^{-2}$ occur less frequently. This is because at larger $N_d$, thus smaller droplets, the occurrence of light rain is suppressed. Cloud droplets must grow larger, leading to deeper clouds in order to reach the size that they can start to precipitate. This leads to a shift in the LWP distribution to the higher values inside the plume. In turn, these deeper clouds in which drops have more water available for growth produce heavier precipitation. Therefore there is an enhancement in the probability of heavy rain (RWP larger than about 180 g m$^{-2}$ ), compared to no-volcano simulation inside of volcano plume, even if the average RWP (Table 1) does not change significantly.

*3. First of all, I appreciate the authors' effort to conduct the numerical simulation by using ICON–NWP and to develop the method for implementing aerosol effects. The effects of aerosols on the cloud microphysical properties can be calculated by the coupling method used in this study. However, the method cannot implement feedback of cloud process to aerosol field. If the authors conducted the ICON–NWP coupled with aerosol transport model or chemical transport model online, the results about the LWP adjustment and cloud fraction adjustment would be changed. I understand that the simulation by ICON–NWP coupled with aerosol transport model or chemical transport model online is one of the future study of authors' group, however, discussions about the*

*limitation about the method used in this study and discussions about the difference of the results from the online coupled model and this study should be added.*

We appreciate the reviewer's suggestion in pointing out the limitation and potential difference of the method used in this study compared to using an interactive aerosol model. The approach used in this study to implement aerosol effects in the ICON-NWP does not consider transport and transformation of aerosols in the model domain and indeed an atmospheric model with interactive aerosol physics would be able to represent the evolution of the aerosol field in more detail. However, an important buffering mechanism, namely the consumption of CCN on activation, is considered in the method used in this study. So CCN are lost when they are activated but will eventually relax back to their initial profile . This was indeed not made clear in the first version and is now clarified in the revised manuscript. It should also be noted that there is an important advantage of our method compared to a fully interactive aerosol scheme, which is that the location of the plume is derived from observations and so is at the same region as in satellite retrievals. This allowed us to analyze inside and outside of the plume in simulations and satellite products with confidence.

*Lines (158-164):*

The approach used in this study does not use an interactive aerosol physics which would simulate the evolution of aerosol field by transport and transformation. However, an important buffering mechanism, namely the consumption of CCN by activation, is considered in this study (Costa-Surós et al.,2020). So the CCN are depleted when they are activated and thereafter are relaxed back to their initial profile. It should also be noted that there is an important advantage of our method compared to a fully interactive aerosol scheme, which is that the location of the plume is derived from observations and therefore is at the same region as in satellite retrievals. This allowed us to analyze inside and outside of the plume in simulations and satellite products with confidence.

**Specific Comments:**

1. *Figure 1: What does the color of Fig. 1 mean? Elevation from sea surface? The caption about the color and the color bar should be added.*

This Figure indicates the domain of the simulation with orography color-coded. Blue color indicates the ocean surface and other colors indicate the elevation of land above sea surface. A color bar is added, and the caption is revised to include information about the color coding.

Figure 1 in the revised manuscript.

Blue color indicates the ocean and color bar indicates the elevation of land above sea surface (in m) in ICON-NWP model.

*2. Line 59: The information of the layer thickness is required.*

We use 75 vertical levels spanning from the surface to 30 km altitude with a vertical resolution of 20 m at the lowest model level that gradually gets coarser towards the model top; the coarsest vertical resolution is 400 m. This information is added to the manuscript.

*Lines (75-76):*

Vertically, 75 layers with top height at 30 km were chosen. The vertical resolution increases towards model top with a model layer thickness of 20 m in the boundary layer and a maximum layer thickens of 400 m near model top.

*3. Line 65–66: What physical variables were used for the initial and boundary condition? Such information is important for other scientists to trace the simulations by other models.*

The physical variables that are used are temperature, horizontal wind components, surface pressure, surface geopotential, geopotential, specific humidity, cloud liquid water content, cloud ice content, rainwater content, snow water content, snow temperature, water content of snow, density of snow, snow albedo, skin temperature, sea surface temperature, soil temperature level 1,2,3,4 (level 1 to 4 are located at 3.5cm, 17.5cm, 49.5cm and 64cm respectively), sea-ice cover, water content of interception storage, surface roughness, Land/sea mask, soil moisture index layer 1,2,3,4. These information are added in the manuscript in a more summarized form.

*Lines (87-88):*

Variables such as temperature, wind, geopotential, humidity and hydrometeors were used in initial and boundary conditions.

*4. Line 68: In this part, the authors indicate that analyses period is from 1 to 7 September, 2014. Is the period corresponding to the period of the calculation? If so, did the author check the effects of spin-up was sufficiently small? In such regional scale simulation, we do not analyze first several hours to avoid the artificial wave generated during initial shock.*

The period of the analysis is from 1 to 7 September 2014. In order to consider the spin-up effect the first 9 hours of simulation were not considered in the analysis.

*Lines (92-93):*

The first 9 hours of the simulations were excluded from analyses so that the spin-up effect is sufficiently small in our simulations.

*5. Line 79: In this part, the authors indicate that the input variables of COSP are temperature, pressure, cloud fraction, and cloud water content. For simulating MODIS's signal, the information of size distribution function is required. The information about the size distribution of hydrometeor should be added in the method.*

Apart from the variables mentioned in the submitted version of manuscript as an example of COSP's inputs, in the MODIS simulator setup which has been used in this study, the effective radius of cloud droplets and ice crystals are used as an input to the simulator. The approach for calculating the effective radius of cloud liquid droplet and ice crystal is to use the actual parameters from the hydrometeor size distribution from the two-moment microphysics scheme in the model. This approach had already been implemented and tested in ICON-NWP by Kretzschmar et al. (2020) and Costa-Surros (2020) for improving cloud radiative properties. Equation B8 in Kretzschmar et al. (2020) indicates how effective radius of liquid cloud droplets and ice crystals are calculated from their respective size distributions. This additional information about the way of using size distribution function as input to the COSP MODIS simulator indeed was lacking and is now added in the revised manuscript.

*Lines (103-107):*

In addition of above mentioned variables, effective radius of liquid cloud droplets and ice crystals are considered as MODIS simulator's input. Effective radius of cloud droplets and ice crystals were calculated from parameters derived from size distribution function of hydrometer in two-moment microphysic scheme. The satellite simulator has previously been implemented and used in ICON-NWP by Kretzschmar et al. (2020).

*6. Section 2.2.: In this section, the authors describe the method for implementing aerosol effects on the ICON–NWP, and the authors shows distribution of column–mean CCN as shown in Fig. 3. I think the distribution of CCN is reasonable. However, there are no information about the vertical distribution of CCN. Based on the body of the manuscript, the data for $SO_2$ was originated from OMPS product. I think that the product is vertical column amount of $SO_2$. Which layer did the authors add the $SO_2$? Based on my experiences, the layer that aerosols are input is really sensitive to the simulated impact of aerosol on cloud microphysical properties. In addition, did the authors assume $SO_2$ gas is as sulfate aerosol particle?*

*As well as the $SO_2$, water vapor is also emitted by the eruption, and the emitted water vapor can affect the meteorological field and cloud properties. Did the author only consider the emission of $SO_2$?*

The activated CCN concentration was computed from aerosol components (including sulfate aerosol) mass mixing ratio in a box model, and this box model requires the sulfate mass mixing ratio at each vertical level. In order to scale the sulfate aerosol from CAMS reanalyses inside the volcanic plume, we scaled the sulfate aerosol mass in each vertical level based on the $SO_2$ retrieved OMPS for the lower troposphere. It is thus assumed that in each level, sulfate aerosols in the troposphere are enhanced in the plume by the same ratio of enhancement as $SO_2$ in the lower troposphere (up to 3 km) in the OMPS satellite retrievals. We now clarified in the revised manuscript that it is not total-column that is used to scale the cloud-active aerosol, but the one in the boundary layer beneath the clouds.

*Lines (181-189):*

*$SO_2$ is considered a proxy of the loading of additional sulfate aerosols in a volcanic plume. The potentially activated CCN concentration was computed from vertically-resolved aerosol components (including sulfate) mass mixing ratio using a box model. The potentially activated CCN profile that is produced to be used as input in ICON-NWP is thus also resolved in vertical levels. In order to define the volcanic plume on the basis of the distribution of sulfate aerosol from the CAMS reanalysis, we scaled each vertical level of sulfate aerosol in CAMS based on the lower troposphere (up to 3 km) column amount of $SO_2$ in OMPS data. In consequence, the vertical distribution within the plume follows the one generated by the reanalysis without the plume, but the scaling makes use of the vertical information from the satellite retrievals in such that only the boundary-layer enhancement is used, i.e. the aerosol that is relevant for the formation of the liquid water clouds investigated in our study.*

*7. Line 193–195: In this part, the authors suggest that the decrease of the probability of clouds with low LWP and the increase of the probability of clouds with high LWP. I agree the suggestion, but I cannot agree "thicker clouds (with high LWP)" and "shallower clouds (with low LWP)" from the results shown the manuscript. Does the word "thick" and "shallow" mean "geometrically thick" and "geometrically thin"? If so, the author should show the cloud geometrical thickness. If the authors just want to discuss the probability of clouds with low and high LWP, the words "thick" and "shallow" are not necessary.*

*As well as the terminology, if the thick clouds increase in the simulation with volcano emission, such difference can result in the change of the circulation. Did the author check the cloud distribution (geographical distribution, vertical structure of clouds and so on)?*

*The discussion about why the decrease of the probability of clouds with low LWP and the increase of the probability of clouds with high LWP occurred is also useful for readers.*

In light of this reviewer's concern, the phrases "thick" and "shallow" were removed from the manuscript because we indeed did not analyze the geometrical thickness of clouds. Rather, by the terms thick and shallow clouds, we meant clouds with higher LWP and lower LWP, respectively. We revised the

manuscript and we added an explanation about the reason for the shift in LWP distribution toward the higher LWP.

*Lines (252-254):*

Considering the simulated profiles, in the simulation with volcano emissions included, there is a decrease in the probability of clouds with lower LWP and an increase in the probability of clouds with higher LWP compared to the no-volcano simulation.

*Lines (271-277):*

In the volcano simulation compared to the no-volcano simulation, for the region inside the plume, RWP of less than about 180 g m$^{-2}$ occur less frequently. This is because at larger $N_d$, thus smaller droplets, the occurrence of light rain is suppressed. Cloud droplets must grow larger, leading to deeper clouds in order to reach the size that they can start to precipitate. This leads to a shift in the LWP distribution to the higher values inside the plume. In turn, these deeper clouds in which drops have more water available for growth produce heavier precipitation. Therefore there is an enhancement in the probability of heavy rain (RWP larger than about 180 g m$^{-2}$ ), compared to no-volcano simulation inside of volcano plume, even if the average RWP (Table 1) does not change significantly.

*Lines (329-332):*

The reason behind this suggestion is when light rain is depressed, cloud droplets must grow deeper in order to reach the size that they can start to precipitate which leads to a shift in the LWP distribution through the higher values, and if cloud droplets grow deeper, they precipitate heavier because they have to fall through more clouds droplets.

*8. In addition, grid line in each frequency distribution (Figs. 4, 5, 6, and 7) will be helpful for readers to distinguish the shift of peak and find the decrease of low LWP and the increase of high LWP.*

The grid lines was added to the figures.

Figure 5,6,7,8 in the manuscript.

*9. Line 218–238: As I mentioned in the general comment, this part is not the main topic of this study. So, I recommend the authors to create new section "discussion" after conclusion or just before the conclusion and move this part to the new section. Again, I understand that the radiative effect is important, but this is not the main topic of this study. Alternatively, I ask the authors to add more descriptions about the enhancement of RWP and precipitation as I mentioned in the general comment.*

The section 'Implications for the radiative impact' is created before the conclusion part and the results about cloud albedo were moved to this section. In addition, in the conclusion, more discussion about the reason for the enhancement of RWP and precipitation as mentioned in the response to the general comment was added.

*Lines (284-310):*

Section 4 : Implications for the radiative impact.

*Lines (329-332):*

The reason behind this suggestion is when light rain is depressed, cloud droplets must grow deeper in order to reach the size that they can start to precipitate which leads to a shift in the LWP distribution through the higher values, and if cloud droplets grow deeper, they precipitate heavier because they have to fall through more clouds droplets.

**Minor Comment:**

*–Line 47: "(Toll et al., 2017)" should be "Toll et al. (2017)"*

Changed.

*Line (47):*

In addition, ambiguous results, with LWP responses of either sign, were obtained by Toll et al. (2017) when analyzing multiple volcanic eruptions

*–Table 1: The unit of LWP and RWP is $g\ m^{-2}$*

Changed.

Table 1 in the manuscript

*–Line 157: I think that "(factual and counterfactual)" is not necessary.*

Changed.

*Lines (68-69):*

The present study aims at a detection and attribution approach, assessing the differences in cloud properties within and outside the volcanic plume by comparing simulations with satellite observations.

*–Figure 4. How did the author define the "Inside" and "Outside" plume?*

The method to define the inside and outside of the plume as is those marine pixels that correspond to the concentration of SO2 more than 1 Dobson in figure 2 are assumed to be located inside the volcano plume and the rest of the pixels are considered as outside of the volcano plume.

*Lines (222-223):*

In order to define the plume, marine pixels that correspond to the $SO_2$ concentrations in the lower troposphere exceeding 1 DU in Figure 2 were chosen.

*–Line 275: The URL of CAMS reanalysis data is not correct. The URL has been moved to "[https://confluence.ecmwf.int/display/COPSRV/Copernicus+Atmosphere+Monitoring+Service+-+CAMS](https://confluence.ecmwf.int/display/COPSRV/Copernicus+Atmosphere+Monitoring+Service+-+CAMS)"*

Corrected.

*Line (350)*

(https://confluence.ecmwf.int/display/COPSRV/Copernicus+Atmosphere+Monitoring+Service+-+CAMS )

**Response to referee comments #2**

*This study performed sensitivity experiments for a region around the North Atlantic to investigate the effects of volcanic smoke aerosols as cloud condensation nuclei (CCN) associated to the eruption in Holuhraun on the cloud properties in the volcanic smoke trails. A scientific goal is to investigate how liquid water path (LWP) and cloud fraction change in response to increase in CCN and subsequently cloud droplet number (Nd) in the volcanic plumes. Regional cloud-resolving simulations with approximately 2.5 km grid spacings were conducted for a week when emission of SO2 from the eruption was clearly identified in satellite observations. The simulation results were compared to satellite observations for cloud to check how the observed difference in cloud properties between in and outside the volcanic plumes was reproduced in the simulations. The simulation considering effects of volcanic smokes replicated the observed increase in Nd in the plumes. However, the same simulation overpredicted increase in significantly LWP and slightly in cloud fraction.*

*I think the current investigation and discussion on the relationship between LWP and Nd or Na (aerosol or CCN number concentration) is insufficient. As described in the current manuscript and in reports from model intercomparison projects (e.g., Quaas et al. 2009), conventional global aerosol transport models tend to overpredict increase in LWP in response to increase in Na or Nd, compared to global-scale satellite observations. However, several recent modeling studies, particularly using high-resolution (cloud- or large-eddy-resolving-scale) models at a regional or global scale, reported little change or even decrease in LWP in the response, according to condition. Their results may be more consistent with the finding in Malavelle et al. (2017), which is the case in this study.*

*First, the manuscript should include careful literature reviews about the advances in recent modeling studies on the sensitivity of LWP to variation in aerosol, CCN, or cloud drop number concentration. Then, more discussion and investigation are needed to examine why the results of the cloud-resolving model simulation in this study contradict findings in some of those recent modeling studies as well as the observation results for the volcanic smoke case. I think at least this effort has to be done toward being acceptable. I have several other major comments listed in the following section. The authors are encouraged to revise the manuscript to improve the quality and readability.*

We thank the reviewer for their assessment of our manuscript. The review helped to improve the

manuscript significantly. In particular, the suggestions led to a clearer formulation of the main messages of the study.

**Major comments:**

*1. LWP-Nd*

*As an example of limited-area large-eddy simulation for aerosol-cloud interaction, Seifert et al. (2015) conducted an extensive series of sensitivity simulations. They reported a negative lifetime effect (unchanged LWP and decrease in cloud cover with increasing Nd) in addition to positive one which has been seen in other previous LES studies, depending on the meteorological condition and the stage of cloud life cycle. Similar dependency of the sensitivity on meteorological condition and cloud regime was found in other LES studies (e.g., Lebo and Feingold 2014). On the other hand, Sato et al. (2018) conducted one-year global cloud-resolving simulation to examine the sensitivity. They successfully reproduced negative λc (the definition can be found in the paper) seen in satellite observations, mostly over regions where cumulus was dominant. They suggested that evaporation process of cloud droplets around cloud top was important to resulting in negative values. More details of the discussion can be found in the paper. As I wrote in the overall comment, since some of other modeling studies could reproduce near-zero and even negative sensitivity, the authors should make efforts to examine and explain why the current simulation could not do it in discussion together with findings in previous studies not limited to those shown above. I understand models have various uncertainty and hence often cannot reproduce observations. But the manuscript should show some advances toward the next step.*

*Seifert, A., Heus, T., Pincus, R., & Stevens, B. (2015). Large-eddy simulation of the transient and near-equilibrium behavior of precipitating shallow convection. Journal of Advances in Modeling Earth Systems, 7(4), 1918-1937.*

*Lebo, Z. J., & Feingold, G. (2014). On the relationship between responses in cloud water and precipitation to changes in aerosol. Atmospheric Chemistry and Physics, 14(21), 11817-11831.*

*Sato, Y., Goto, D., Michibata, T., Suzuki, K., Takemura, T., Tomita, H., & Nakajima, T. (2018). Aerosol effects on cloud water amounts were successfully simulated by a global cloud-system resolving model. Nature Communications, 9(1), 1-7.*

*L143.*

We appreciate the reviewer's suggestion; indeed it is important to provide a balanced and detailed report about the state of the art. It is of course difficult to be exhaustive, but we made an effort to discuss the key results of large-eddy simulations or kilometer-resolution simulations of the effect of enhanced aerosol on LWP for various cloud regimes in the revised manuscript. Among these, the papers the reviewer suggested are now discussed. We made an effort to include a number of further relevant studies, such as the ones by Ackerman et al. (2004) and Bretherton et al. (2007) that also

described the relevant processes in stratocumulus, where reduced sedimentation of smaller droplets may lead to a decrease of the cloud-top entrainment rate and thus a decrease of LWP. The reasons for discrepancies in different cloud-resolving modeling studies are now also discussed in the revised manuscript.

*Lines (48-63):*

Several cloud-resolving modeling studies on the sensitivity of LWP to variation in cloud droplet number concentration have been conducted. Bretherton et al. (2007) examined the effect of entertainment and sedimentation rate on LWP in stratocumulus cloud regimes using large eddy simulation (LES). Their results explained the process details of the conclusions by Ackerman et al. (2004), namely that sedimentation leads to decrease of entrainment rates and increase of LWP. Seifert et al. (2015) conducted a set of LES simulations over fair weather cumulus cloud regimes over the subtropical ocean. They concluded that in this cloud regime, the response of LWP on enhancing $N_d$ was almost negligible in equilibrium, and slight reduction in cloud cover was obtained, leading to a negative cloud lifetime effect, compensating the positive radiative forcing of the Twomey effect. Lebo and Feingold (2014) performed LES simulations of two different cloud regimes of marine stratocumulus and trade wind cumulus clouds. They showed different relationships between relative LWP response to relative change of aerosol concentration $N_a$ , a term the called λ, and the precipitation probability susceptibility (SPOP). For trade wind cumulus clouds regime, λ decreases with enhancement of $S_{POP}$ , because of entrainment and evaporation rate effect in cumulus clouds. In stratoscumulus clouds, λ and $S_{POP}$, in contrast, were positively related. In this case, aerosol-induced evaporation–entrainment and/or sedimentation–entrainment effects restricted further increase in LWP in their simulations. Sato et al. (2018) conducted one-year global cloud-resolving simulation to examine the sensitivity of liquid water content (LWC) to aerosol loading. They demonstrated that in their model, the condensation process in the lower part of clouds is associated with positive LWC response and evaporation process in upper part of clouds is responsible for negative response to additional aerosols loading.

**2. Meteorological and cloud information of the target case**

*The manuscript should show what meteorological condition and what types of cloud were dominant in the period and the domain for the simulations. These information is quite important in the discussion because previous studies, e.g., in comment #1, showed some dependency of the aerosol–cloud interaction on those factors. Some MODIS true–color images may help it. And another question, is only warm–topped cloud with cloud top temperature over 273.15 K analyzed and is the other cold–topped cloud excluded?*

The reviewer is right that this is key information and indeed the broad idea of synoptically driven clouds is insufficient. We now include a new Figure 1 which shows the MODIS visible images from 1

to 7 September 2014 over the simulation domain. A synoptic frontal system is dominant over the North Atlantic ocean. The cloud regime consists of both ice and liquid phase clouds and remains approximately the similar regime during the simulation period. Malavelle et al. (2017) analyzed the cloud regime  derived from satellite measurement and showed that the region surrounding the Holuhraun volcano contains the whole spectrum of liquid-dominated cloud regimes. In our analysis, we exclude clouds over land both in satellite observation and simulations. The method to select liquid phase clouds in the MODIS product was to use cloud phase optical properties. This product contains a flag dedicated to liquid phase clouds, therefore, all the grid points with liquid phase clouds were chosen for our analysis. For simulations, the COSP simulator computes the microphysical properties (effective radius, liquid water path, and cloud optical depth) for the liquid and ice phase clouds separately and we used the simulator output dedicated to the liquid phase.

[Figure]

Figure 1. Visible image of MODIS-AQUA from 1 to 7 September 2014.

*Lines (219-205):*

 It has been shown that meteorological conditions and cloud regimes are important to determine the effect of additional aerosol loading on cloud microphysical properties. Figure 4 indicates visible image obtained form MODIS-AQUA satellite retrievals. A synoptic frontal system is located over the North Atlantic ocean and contains large-scale, mostly stratiform ice and liquid phase clouds. These conditions remain similar during the simulation period. In order to to select liquid phased clouds in the MODIS data, the Cloud Phase Optical Properties flag was used. For simulations, the COSP simulator produces the microphysical properties for the liquid and ice phase clouds separately and we used only the outputs dedicated to liquid phase clouds in our analyses.

Figure 4 in the revised manuscript.

**3. Vertical distribution of the volcanic aerosol plume**

*The OMPS satellite retrieval products were used to identify the column total SO2, and then sulfate aerosol mass mixing ratio was calculated based on the difference in column total SO2 between in and outside the volcanic (around Ln. 143). But I think the vertical profiles of SO2 and sulfate aerosol concentrations might differ between, because they might be contaminated in limited vertical layers into which smoke was injected. How did the authors consider the vertical injection or vertical distribution of the volcanic aerosol plume? Or, maybe I am confused, does the model not need the information of vertical distribution of aerosol but just use column–integrated value to calculate activated CCN concentration at each vertical level?*

It seems that we did not explain very clearly what we did to define the plume and in light of the reviewer's comment we now clarified and enhanced the corresponding section in the revised manuscript. The concentration of potentially activated CCN was computed from aerosol components (including sulfate aerosol) mass mixing ratio using a box model, and this box model requires the mass mixing ratio at each level. The potentially activated CCN profile, produced to be used as input in ICON is vertically resolved. In order to define the volcanic plume on the basis of the distribution of sulfate aerosol from the CAMS reanalysis, we scaled each vertical level of sulfate aerosol in CAMS based on the lower troposphere column amount of $SO_2$ in OMPS data. So we assume that in each level sulfate aerosols in the troposphere are enhanced in the plume as the same ratio of column amount of $SO_2$ in the lower troposphere (up to 3 km) in OMPS data. In consequence, the vertical distribution within the plume follows the one generated by the reanalysis without the plume, but the scaling makes use of the vertical information from the satellite retrievals such that only the boundary-layer enhancement is used, i.e. the aerosol that is relevant for the formation of the liquid-water clouds investigated in our study.

*Lines (181-189):*

  *SO$_2$ is considered a proxy of the loading of additional sulfate aerosols in a volcanic plume. The potentially activated CCN concentration was computed from vertically-resolved aerosol components (including sulfate) mass mixing ratio using a box model. The potentially activated CCN profile that is produced to be used as input in ICON-NWP is thus also resolved in vertical levels. In order to define the volcanic plume on the basis of the distribution of sulfate aerosol from the CAMS reanalysis, we scaled each vertical level of sulfate aerosol in CAMS based on the lower troposphere (up to 3 km) column amount of SO$_2$ in OMPS data. In consequence, the vertical distribution within the plume follows the one generated by the reanalysis without the plume, but the scaling makes use of the vertical information from the satellite retrievals in such that only the boundary-layer enhancement is used, i.e. the aerosol that is relevant for the formation of the liquid water clouds investigated in our study.*

**4. Definitions of LWP in MODIS product and the simulation**

*It is clearly written that Nd in the simulations were calculated using a satellite simulator through same pathway as for the MODIS products. But what about LWP? The definition of LWP has large uncertainty between the satellite products and the model simulation even using a simulator because bulk cloud microphysics has a category gap between cloud water and rain. This is problem in the radiative transfer calculation in simulator to determine LWP that is consistent with that in satellite products. This problem may affect the calculation of other variables such as Nd also.*

We appreciate the reviewer's concern about the category gap that exists between cloud water and rain in bulk microphysics schemes. We add this point that the category gap between cloud droplets and rain in size distribution because of using a bulk microphysics scheme in the simulation could cause some uncertainty in LWP between simulations and MODIS and this issue could also affect the computation of cloud droplet number concentration. However, the issue is in our opinion less important when analyzing the differences between the clouds within and outside the volcanic plume.

 *Lines (113-117) :*

 It should be mentioned that even using a satellite simulator there is an uncertainty between definition of LWP in simulation and MODIS observations, because the bulk microphysic scheme has gap in size distribution between cloud droplets and rain that is not necessarily the same as in the visible/near-infrared retrievals by MODIS. To a lesser extent, this issue may also affect the computation of N$_d$ .

**5. Discussion on cloud fraction**

*I think 2.5 km model grid spacing may be still coarse for comparative discussion of cloud fraction over ocean with the Level-2 MODIS-Aqua cloud product (swath 1km). The model simulation*

*might miss parts of scattered shallow cumulus over the ocean and overemphasize extent of deeper cloud. This might contribute the overprediction of positive cloud lifetime effects on cloud fraction in the plumes in the simulation too. The shallow convection parameterization of Tiedtke (1989) has no effects on the calculation of the cloud fraction, correct?*

In the MODIS-Aqua level 2 cloud fraction product is available at 5x5 km pixels . While of course finer resolution can improve the results in comparing the resolution of simulation to MODIS, it can be seen that this resolution is sufficient for the purpose of discussing  the results in this study. We used a grid-scale cloud cover scheme in our simulation. This scheme works in a way that if the sum of specific cloud water content and specific cloud ice content is larger than a threshold (1e-8 kg/kg), cloud fraction is set to 1 and else set to 0. The Tiedtke (1989) shallow convection implicitly contributes to the computation of specific cloud water and ice content but doesn't have an explicit effect in the computation of cloud fraction. These explanations are now added to the revised manuscript.

*Lines (80-84):*

A grid-scale cloud cover scheme was employed in simulations. In this scheme, if the sum of specific cloud water content and specific cloud ice content is larger than a certain threshold, cloud fraction is set to 1 and else set to 0. Therefore, the Tiedtke (1989) shallow convection scheme implicitly contributes to the computation of specific cloud water and ice content but does not have an explicit effect in the computation of cloud fraction.

*Lines (111-112):*

swaths with 1 km spatial resolution for $r_e$ , cloud optical thickness ($\tau_c$ ), LWP along with cloud fraction with 5 km spatial resolution were used and remapped to the model resolution to have an accurate comparison.

**6. CERES 20 km resolution**

*Is the 20 km resolution of the CERES products enough to distinguish in and outside the smoke plume? The spatial scales of the smoke trails are unclear to me. And what algorithm was used for remapping the model results from the native model grid structure to those with 20 km grid spacing? The selection of the algorithm may strongly affect the results because it was from fine to very coarse grid structures.*

Indeed it is evident that more detail is needed in this regard. Grid-points with $SO_2$ concentrations in the lower troposphere exceeding 1DU are considered to constitute the plume, and $SO_2$ concentration was obtained from OMPS satellite retrievals which are in 50km×50km footprint data in Level 2. We remapped the Level 2 data into the 50km (0.5° degree) resolution, and due to the fact that CERES products are on 20km resolution, it has sufficient resolution to identify the plume. We used the CDO (climate data operators) tools to remap the original unstructured ICON grid to a regular latitudelongitude grid. The same algorithm was used to remap the model grid with 2.5km resolution to 20km resolution to be compared with the CERES product. This CDO module contains operators for an inverse distance weighted average remapping of the four nearest neighbor values of fields between grids in spherical coordinates. Figure 2 shows a comparison between net-shortwave-flux at the top of atmosphere from the simulation which is remapped by the mentioned operator to both 2.5km and 20km resolution. The overall pattern and values are very similar even though there are some small differences due to the remapping from fine to coarse resolution. The relevant additional information is added to the revised manuscript.

[Figure]

Figure 2 . A comparison between the net shortwave flux of different resolution
of 2.5 km and 20km.

*Lines (287-290):*

For the comparison, the simulation output was remapped by distance weighted average remapping of the four nearest neighbor values method to 20 km horizontal resolution to be consistent with the resolution of the CERES footprint.

*Line(290-293):*

Grid points with SO2 concentrations in the lower troposphere exceeding 1 DU are considered to constitute the plume, and SO2 concentration was obtained from OMPS satellite retrievals which are in 50 km footprint data in level-2. We remapped the level-2 data into the 50 km resolution, and due to the fact that CERES products in 20 km resolution, it has sufficient resolution to identify the plume.

**Minor comments:**

*Ln. 39: "cloud" => "could"*

Corrected.

*Line (39):*

*the overall changes in the effective radiative forcing could be minor on larger scales (Khain et al., 2008; Stevens and Feingold, 2009).*

*Ln. 84: Same question as in major comment #6, what algorithm was used for remapping?*

The MODIS Level 2 data at 1km resolution was remapped to latitude-longitude grids with 2.5 km resolution. The MODIS swaths were used and for each specific point, the mean value of different amounts of a specific variable in each swath were computed.

*Lines (112-113):*

*To remap the MODIS granule to a latitude / longitude grid, for each specific point the mean value of each variable in each swath is computed.*

*Around Ln. 110: Can you summarize the variables in the look–up table and the value ranges into a table?*

The look-up table consists of potentially activated CCN number concentrations for 10 specific vertical velocities and height for each hybrid-sigma-pressure level (60 levels) and 3 hour interval. This lookup table was calculated offline for 1 to 7 September 2014 (the period of simulation). Here, to show the range of values, we choose 2 September and compute a daily mean. The model level corresponding to approximately (850 hPa) was chosen. The table below summarizes each specific vertical velocity that has been used in the box model for computations of potentially activated CCN concentration. The value range is shown as the mean value for the whole domain and the first and the third quartile of grid point values.

| 2 September 2014 | | | |
|---|---|---|---|
| Variables | First Quartile($cm^{-3}$) | Mean value($cm^{-3}$) | Third Quartile($cm^{-3}$) |
| CCN_act (w = 0.01 m/s ) | 5 | 7 | 8 |
| CCN_act (w = 0.0278 m/s) | 20 | 26 | 32 |
| CCN_act (w = 0.0774 m/s ) | 54 | 73 | 87 |
| CCN_act (w = 0.215 m/s) | 117 | 166 | 204 |
| CCN_act (w = 0.599  m/s) | 230 | 334 | 414 |
| CCN_act (w =  1.67 m/s) | 406 | 605 | 753 |
| CCN_act (w = 4.64  m/s) | 639 | 994 | 1235 |
| CCN_act (w =  12.9 m/s) | 918 | 1492 | 1842 |
| CCN_act (w =  35.9 m/s) | 1219 | 2070 | 2534 |
| CCN_act (w = 100 m/s) | 1528 | 2691 | 3271 |

*Lines (353-360):*

**Appendix A: Look-up table of potentially activated CCN number concentrations**

The look-up table consists of potentially activated CCN number concentrations for 10 specific vertical velocities and height for each hybrid-sigma-pressure level (60 levels) and 3 hour interval. This look-up table was calculated offline for 1 to 7 September 2014 (the period of simulation). In order to show the range of values, we choose 2 September and compute its daily mean. The model level corresponding to approximately (850 hPa) was chosen. The table A1 summarizes each specific vertical velocity that has been used in the box model for computations of potentially activated CCN concentration. The value range is shown as the mean value for the whole domain and the first and the third quartile of grid point values.

Table A1 in the revised manuscript.

*Table 1: Could you add comparison of τc and re into Table 1 too?*
The variables effective radius and cloud optical thickness are added to Table 1. It can be seen that effective radius decreased inside the plume by 7% compared to outside the plume. In the MODIS data, the effective radius decreased by 8% inside the plume compared to outside the plume. Cloud optical thickness increased by 33% inside the plume compared to outside the plume in MODIS. This

enhancement is about 24% in the "volcano" simulation while in the "no-volcano" simulation, cloud optical thickness decreased by 3% inside the plume compared to outside the plume. This discussion is now added to the manuscript.

*Lines (241-247):*

Table 1 further lists the mean values and changes for re and $\tau_c$. The effective radius decreased inside of plume by 7 % compare to outside the plume in volcano simulation. In the no-volcano simulation, there is no difference in re inside vs. outside the plume. In the MODIS retrievals, $r_e$ decreased by 8 % inside the plume compared to outside the plume, very similar to the change in the simulation. This is consistent with the agreement in plume enhancement for $N_d$ and despite the simulated change in LWP. Also the cloud optical thickness shows a consistent increase in MODIS as in the volcano simulation, whereas the no-volcano simulation shows a (very slight) decrease in $\tau_c$ inside the plume.

*Figs.2 and 3: please add lines of latitude and longitude*
The grid lines were added to the figures.
Figures 2 and 3 in the manuscript

*Ln. 253-255: These sentences are a bit awkward. Please rephase and improve the readability.*

The sentence is rephrased.

*Lines (324-327):*

However, the mean increase in MODIS is very close to the result of the no-volcano run. This almost zero enhancement in MODIS on average is because of decrease in LWP for the clouds with low LWP, and an enhancement of LWP for large LWP values which is consistent with the results of the ICON-NWP model, nevertheless, the model, exaggerates the increase in large LWP values.

*Ln. 256-257: The sentence is confusing. The vertical axis of the plots in Fig. 6 is at a log-scale. The frequency of high RWP over 200 gm-2 in the volcano simulation is quite or neglectably small, and the difference in mean RWP in the plumes is due to the difference in the frequency of lower RWP values.*

The sentence is rephrased.

*Lines (327-329):*

In the model the reason for the enhancement of LWP in the volcano simulation was the decrease in precipitation compared to no-volcano simulation by 15 % on average, due to a decrease in light rain in volcano simulation compared to no-volcano simulation.

---

## Referee Report (RR1)

Review of the manuscript numbered acp-2022-38, revised version 1

Title: "Impact of Holuhraun volcano aerosols on clouds in cloud-system resolving simulations" written by Haghighatnasab et al.

Manuscript number: "acp-2022-38".

Decision: "Minor revision"

The authors have done a good job responding to most of my concerns and comments. I thankful for the authors' efforts to revise the manuscript. However, a few comments were not addressed. So, my decision is minor revisions.

The number of lines, sections, and comments written in blue characters, mean my comments in previous round of review.

**The comments that have not been addressed:**

Specific comment for Section 2.2.: In this section, the authors describe the method for implementing aerosol effects on the ICON-NWP, and the authors shows distribution of column-mean CCN as shown in Fig. 3. I think the distribution of CCN is reasonable. However, there are no information about the vertical distribution of CCN. Based on the body of the manuscript, the data for $SO_2$ was originated from OMPS product. I think that the product is vertical column amount of $SO_2$. Which layer did the authors add the $SO_2$? Based on my experiences, the layer that aerosols are input is really sensitive to the simulated impact of aerosol on cloud microphysical properties. In addition, did the authors assume $SO_2$ gas is as sulfate aerosol particle?

The authors added some descriptions about the treatment of the $SO_2$ emitted from volcano (Line 185-193 of the revised manuscript). However, it is not clear for me about the treatment of $SO_2$.

Based on the revised manuscript, the authors added $SO_2$ retrieved from OMPS data product with a "scaling" to lower troposphere (i.e., up to 3 km height). However, how did the authors "scale" the data? Was the $SO_2$ added uniformly up to 3 km height? or added some vertical distribution (i.e., decreased exponentially with height)? The author should add more detailed information. The figure of the vertical profile of the activated CCN in supplemental material will be helpful for the readers.

In addition, I'm not sure about the treatment of aerosols in the model. In the line 164-165, the authors indicate that the consumption (or depletion) of CCN can be considered in the method used in this study. However, the method in this study used observation of OMPS for volcanic $SO_2$ as an external data, and the consumption and depletion process of aerosol cannot be considered in this

method. In addition, the authors refer a literature of Costa-Suros et al. (2020), but in my understanding, Costa-suros et al. (2020) used offline aerosol transport model. If the authors used offline aerosol model, the consumption and depletion process can be calculated explicitly as a wet deposition process. However, I'm wondering the consumption process can be included by the method in this study that is described in Section 2.2.

If I misunderstand the method used in this study, please explain the method more clearly.

In addition, I cannot find the answer from the authors to my comment:

Specific comment for Section 2.2.: As well as the $SO_2$, water vapor is also emitted by the eruption, and the emitted water vapor can affect the meteorological field and cloud properties. Did the author only consider the emission of $SO_2$?

Line 157: I think that "(factual and counterfactual)" is not necessary.: The word, "factual" and "counterfactual" are remained in conclusion in revised manuscript.

**Technical comments:**

Figure 4: Information of data source of these satellite image should be included in the caption (I know the information is included in acknowledgement, but I think the information should be added in the caption).

Line 145: If the authors add the literature of Sato et al. (2018), which was used NICAM, Goto et al. (2020, GMD, doi:10.5194/gmd-13-3731-2020) can be added as an example of the model (NICAM) using ARG-parameterization.

**Additional comment**

Line 334-335: In this part, the authors suggest that the model exaggerates the increase in large LWP values. Based on my experience, such exaggeration commonly occurs in the model, in which effects of clouds are calculated by cloud microphysical model. Why does such exaggeration occur? If the authors have any answer or some speculation, some comments about this exaggeration will helpful for scientific community.

---

## Author Response (AR2)

**Response to the Reviewer's comments version 2**

*We would like to thank the reviewers for their effort in helping us improve the manuscript. Below we respond point-by-point to the comments, with the reviewer comments in black, our responses in black, and the changes in the revised manuscript in blue. The line numbers are for the revised manuscript version.*

**Response to referee comments #1**

*The authors have done a good job responding to most of my concerns and comments. I thankful for the authors' efforts to revise the manuscript. However, a few comments were not addressed. So, my decision is minor revisions.*

We thank the reviewer for their assessment of our manuscript. The review helped to improve the manuscript significantly.

*The number of lines, sections, and comments written in blue characters, mean my comments in previous round of review.*

*The comments that have not been addressed:*

*Specific comment for Section 2.2.: In this section, the authors describe the method for implementing aerosol effects on the ICON–NWP, and the authors shows distribution of column–mean CCN as shown in Fig. 3. I think the distribution of CCN is reasonable. However, there are no information about the vertical distribution of CCN. Based on the body of the manuscript, the data for $SO_2$ was originated from OMPS product. I think that the product is vertical column amount of $SO_2$. Which layer did the authors add the $SO_2$? Based on my experiences, the layer that aerosols are input is really sensitive to the simulated impact of aerosol on cloud microphysical properties. In addition, did the authors assume $SO_2$ gas is as sulfate aerosol particle?*

*The authors added some descriptions about the treatment of the $SO_2$ emitted from volcano (Line 185–193 of the revised manuscript). However, it is not clear for me about the treatment of $SO_2$. Based on the revised manuscript, the authors added $SO_2$ retrieved from OMPS data product with a "scaling" to lower troposphere (i.e., up to 3 km height). However, how did the authors "scale" the data? Was the $SO_2$ added uniformly up to 3 km height? or added some vertical distribution (i.e., decreased exponentially with height)? The author should add more detailed information. The figure of the vertical profile of the activated CCN in supplemental material will be helpful for the readers.*

We thank the reviewer for the suggestion. The scaling of CCN was done by computing the distribution of scaling based on the enhancement of SO2 inside the plume relative to the mean SO2 value outside of the plume in the lower troposphere (up to 3km). Then the sulfate concentrations in the CAMS reanalyses inside of the plume were scaled at each level by the computed ratio. So the sulfate aerosol concentration at each level was scaled with the same ratio but the concentration of the sulfate is not the same at each level because the background concentration is different at each level. In the next step, the box model was employed on the scaled sulfate aerosol concentration, and the scaled CCN profile was obtained. To clarify the vertical distribution of CCN more specifically, we add figure A1 in Appendix in the manuscript depicting the mean value of column SO2 concentration inside of plume in the lower, middle, and upper troposphere in OMPS retrievals along with the vertical profile of mean CCN concentration inside of the plume in the no-volcano and volcano run and outside of the plume in the no-volcano run for one specific vertical velocity (0.559 m/s) on 2 September 2014.

[Figure]

Line(362):
**Vertical profile of activated CCN**
The scaling of CCN was done by computing the distribution of scaling based on the enhancement of SO2 inside the plume relative to the mean SO2 value outside of the plume in the lower troposphere (up to 3km). Then the sulfate concentrations in the CAMS reanalyses inside of plume were scaled at each level by the computed ratio. So the sulfate aerosol concentration at each level was scaled with the same ratio but the concentration of the sulfate is not the same at each level because the background concentration is different at each level. In the next step, the box model was employed on the scaled sulfate aerosol

concentration, and the scaled CCN profile was obtained. To determine the vertical distribution of CCN more specifically, figure A1 shows the mean value of column SO2 concentration inside of the plume in the lower, middle, and upper troposphere in OPMS retrievals along with the vertical profile of mean CCN concentration inside of the plume in the no-volcano and volcano run and outside of the plume in the no-volcano run for one specific vertical velocity (0.559 m/s) on 2 September 2014.
Figure A1 in the manuscript.

*In addition, I'm not sure about the treatment of aerosols in the model. In lines 164-165, the authors indicate that the consumption (or depletion) of CCN can be considered in the method used in this study. However, the method in this study used observation of OMPS for volcanic SO2 as external data, and the consumption and depletion process of aerosol cannot be considered in this method. In addition, the authors refer a literature of Costa-Suros et al. (2020), but in my understanding, Costa-suros et al. (2020) used offline aerosol transport model. If the authors used*
*offline aerosol model, the consumption and depletion process can be calculated explicitly as a wet deposition process. However, I'm wondering the consumption process can be included by the method in this study that is described in Section 2.2.*
*If I misunderstand the method used in this study, please explain the method more clearly.*

We added an additional sentence to now clarify how exactly this is done: "This is implemented by a simple prognostic equation for the CCN concentration that considers a sink for CCN at droplet activation and a source by relaxation to the prescribed CCN profile, advection is not computed."

Line(161-162):
This is implemented by a simple prognostic equation for the CCN concentration that considers a sink for CCN at droplet activation and a source by relaxation to the prescribed CCN profile, advection is not computed.

*In addition, I cannot find the answer from the authors to my comment:*
*Specific comment for Section 2.2.: As well as the SO2, water vapor is also emitted by the eruption, and the emitted water vapor can affect the meteorological field and cloud properties. Did the author only consider the emission of SO2?*

In the method used in this study, only emissions of SO2 were considered and the emission of water vapor from the volcano hasn't been taken into the account. We add this point to the manuscript to clarify this point.

Line (190):
It should be mentioned that in this study, the emission of water vapor from volcanic eruption hasn't been taken to account.

*Line 157: I think that "(factual and counterfactual)" is not necessary.: The word, "factual" and "counterfactual" are remained in conclusion in revised manuscript.*

The phrases "(factual and counterfactual)" are removed.

*Technical comments:*
*Figure 4: Information of data source of these satellite image should be included in the caption (I know the information is included in acknowledgement, but I think the information should be added in the caption).*

Thank you so much for pointing it out. The caption is revised to add the source of satellite images.

*Line 145: If the authors add the literature of Sato et al. (2018), which was used NICAM, Goto et al. (2020, GMD, doi:10.5194/gmd-13-3731-2020) can be added as an example of the model (NICAM) using ARG-parameterization.*

The paper is cited in the manuscript as an example of ARG-parametrization.

*Additional comment*
*Line 334-335: In this part, the authors suggest that the model exaggerates the increase in large LWP values. Based on my experience, such exaggeration commonly occurs in the model, in which the effects of clouds are calculated by cloud microphysical model. Why does such exaggeration occur? If the authors have any answer or some speculation, some comments about this exaggeration will helpful for scientific community.*

One needs to dig deeper into processes in the microphysics scheme to be able to explain in detail why this exaggeration in the enhancement of LWp has occurred, which this investigation has not been done in this study. But we can mention some hypotheses, for example, the finer horizontal resolution may help to improve the estimated enhancement in LWP. In addition, modifying the autoconversion rate in the microphysics scheme could lead to a better estimate of the suppression of light rain (drizzles).

*Response to referee comments #2*

*The authors sufficiently addressed my previous review comments. I think the manuscript is acceptable once the following very minor points are revised. The authors do not need to make responses to the comments.*

We thank the reviewer for their assessment of our manuscript. The review helped to improve the manuscript significantly.

*Minor comments:*
*Ln. 81–83. The sentence is a little bit confusing. I suggest simply saying, "The Tiedtke (1989) shallow convection scheme contributes to the computation of specific cloud water and ice content.", if my understanding is correct.*

The sentence is now revised consistent with what is suggested.

Lines(81-82): the Tiedtke (1989) shallow convection scheme contributes to the computation of specific cloud water and ice content.

*Ln. 245: "and despite the simulated change in LWP" I think this phrase is unnecessary here in because the discussion on LWP starts from the next paragraph.*

The phrase is removed.

Line (248)

Line
*Ln. 262: ")" should be after "6"*

Revised

Line (263)

*Section 5 Conclusions: Some sentences are awkward.*

We edit some sentences in the conclusion part in order to improve readability.

Line (313): Conclusion section.